# Inferring synaptic inputs from spikes with a conductance-based neural encoding model

**Kenneth W Latimer[1][†]\*, Fred Rieke[1], Jonathan W Pillow[2]\***

[1]Department of Physiology and Biophysics, University of Washington, Seattle, United States; [2]Princeton Neuroscience Institute, Department of Psychology, Princeton University, Princeton, United States

**Abstract** Descriptive statistical models of neural responses generally aim to characterize the mapping from stimuli to spike responses while ignoring biophysical details of the encoding process. Here, we introduce an alternative approach, the conductance-based encoding model (CBEM), which describes a mapping from stimuli to excitatory and inhibitory synaptic conductances governing the dynamics of sub-threshold membrane potential. Remarkably, we show that the CBEM can be fit to extracellular spike train data and then used to predict excitatory and inhibitory synaptic currents. We validate these predictions with intracellular recordings from macaque retinal ganglion cells. Moreover, we offer a novel quasi-biophysical interpretation of the Poisson generalized linear model (GLM) as a special case of the CBEM in which excitation and inhibition are perfectly balanced. This work forges a new link between statistical and biophysical models of neural encoding and sheds new light on the biophysical variables that underlie spiking in the early visual pathway.

**\*For correspondence:**
latimerk@uchicago.edu (KWL);
pillow@princeton.edu (JWP)

**Present address:** [†]Department of Neurobiology, University of Chicago, Chicago, United States

## Introduction

Studies of neural coding seek to reveal how sensory information is encoded in neural spike responses. A complete understanding this code requires knowledge of the statistical relationship between stimuli and spike trains, as well as the biophysical mechanisms by which this transformation is carried out. A popular approach to the neural coding problem involves 'cascade' models, such as the linear-nonlinear-Poisson (LNP) or generalized linear model (GLM), to characterize how external stimuli are converted to spike trains. These descriptive statistical models describe the encoding process in terms of a series of stages: linear filtering, nonlinear transformation, and ending with noisy or conditionally Poisson spiking (*Chichilnisky, 2001*; *Paninski, 2004*; *Vintch et al., 2012*; *Park et al., 2013*; *Theis et al., 2013*; *Vintch et al., 2015*). These models have found broad application to neural data, and the Poisson GLM in particular has provided a powerful tool for characterizing neural encoding in a variety of sensory, cognitive, and motor brain areas (*Harris et al., 2003*; *Truccolo et al., 2005*; *Pillow et al., 2008*; *Gerwinn, 2010*; *Stevenson et al., 2012*; *Weber et al., 2012*; *Park et al., 2014*; *Hardcastle et al., 2015*; *Yates et al., 2017*).

However, there is a substantial gap between cascade-style descriptive statistical models and mechanistic or biophysically interpretable models. In real neurons, stimulus integration is nonlinear, arising from an interplay between excitatory and inhibitory synaptic inputs that depend nonlinearly on the stimulus; these inputs in turn drive conductance changes that alter the nonlinear dynamics governing membrane potential. In retina and other sensory areas, the tuning of excitatory and inhibitory inputs can differ substantially (*Roska et al., 2006*; *Trong and Rieke, 2008*; *Poo and Isaacson, 2009*; *Cafaro and Rieke, 2013*), meaning that a single linear filter is not sufficient to describe stimulus integration in single neurons. Determining how stimuli influence neural conductance changes,

and thus the computations that neurons perform, therefore remains an important challenge. This challenge is exacerbated by the fact that most studies of neural coding rely on extracellular recordings, which detect only spikes and not synaptic conductance changes that drive them.

Here, we aim to narrow the gap between descriptive statistical models and biophysically interpretable models, while remaining within the domain of models that can be estimated from extracellular spike train data (*Meng et al., 2011*; *Meng et al., 2014*; *Volgushev et al., 2015*; *Lankarany, 2017*). We first introduce a quasi-biophysical interpretation of the standard Poisson GLM, which reveals its equivalence to a constrained conductance-based model with equal and opposite excitatory and inhibitory tuning. We then relax these constraints in order to obtain a more flexible and more realistic conductance-based model with independent tuning of excitatory and inhibitory inputs. The resulting model, which we refer to as the *conductance-based encoding model* (CBEM), can capture key features of real neurons such as shunting inhibition and time-varying changes in gain and membrane time constant. We show that the CBEM can predict excitatory and inhibitory synaptic conductances from stimuli and extracellular spike trains alone, which we validate by comparing model predictions to conductances measured with intracellular recordings in macaque parasol and midget retinal ganglion cells (RGCs). This work differs from previous cascade modeling approaches for separating excitatory and inhibitory inputs (e.g., *Butts et al., 2011*; *Ozuysal et al., 2018*; *McFarland et al., 2013*; *Maheswaranathan et al., 2018*) by explicitly defining the model components in a biophysical framework and directly comparing model predictions to measured excitation and inhibition tuning in individual cells. We also show that the CBEM outperforms the standard GLM at predicting retinal spike responses to novel stimuli. These differences highlight the CBEM's ability to shed light on the computations performed by sensory neurons in naturalistic settings.

## Results

### Background: Poisson GLM with spike history

The Poisson GLM provides a simple yet powerful description of the encoding relationship between stimuli and neural responses (*Truccolo et al., 2005*). A recurrent Poisson GLM, often referred to in the neuroscience literature simply as 'the GLM', describes neural encoding in terms of a cascade of linear, nonlinear, and probabilistic spiking stages (*Figure 1a*). The GLM parameters consist of a stimulus filter $\mathbf{k}$, which describes how the neuron integrates an external stimulus, a post-spike filter $\mathbf{h}$, which captures dependencies on spike history, and a baseline $b$ that determines baseline firing rate in the absence of input. The outputs of these filters are summed and passed through a nonlinear function $f_r$ to obtain the conditional intensity for an inhomogeneous Poisson spiking process. The model can be written concisely in discrete time as:

$$\lambda_t = f_r(\mathbf{k} \cdot \mathbf{x}_t + \mathbf{h} \cdot \mathbf{y}_t^{hist} + b) \quad \text{(spike rate)} \tag{1}$$

$$y_t \mid \lambda_t \sim \text{Poiss}(\Delta\lambda_t) \quad \text{(probabilistic spiking)} \tag{2}$$

where $\lambda_t \geq 0$ is the spike rate (or conditional intensity) at time $t$, $\mathbf{x}_t$ is the spatio-temporal stimulus vector at time $t$, $\mathbf{y}_t^{hist}$ is a vector of relevant spike history at time $t$, and $y_t$ is the spike count in bin of size $\Delta$. Although spike generation is conditionally Poisson, the model can capture complex history-dependent response properties such as refractoriness, bursting, bistability, and adaptation (*Weber and Pillow, 2016*). Additional filters can be added to the model in order to incorporate dependencies on covariates of the response such as spiking in other neurons or local field potential recorded on nearby electrodes (*Truccolo et al., 2005*; *Pillow et al., 2008*; *Kelly et al., 2010*). A common choice for the nonlinearity is exponential, $f(z) = \exp(z)$, which corresponds to the 'canonical' inverse link function for Poisson GLMs.

Previous literature has offered a quasi-biological interpretation of the GLM known as 'soft threshold' integrate-and-fire (IF) model (*Plesser and Gerstner, 2000*; *Gerstner, 2001*; *Paninski et al., 2007*; *Mensi et al., 2011*). This interpretation views the summed filter outputs as the neuron's membrane potential. This is similar to the standard IF model in which membrane potential is a linearly filtered version of input current (as opposed to conductance-based input). The nonlinear function $f_r$ can be interpreted as a 'soft threshold' function that governs a smooth increase in the instantaneous

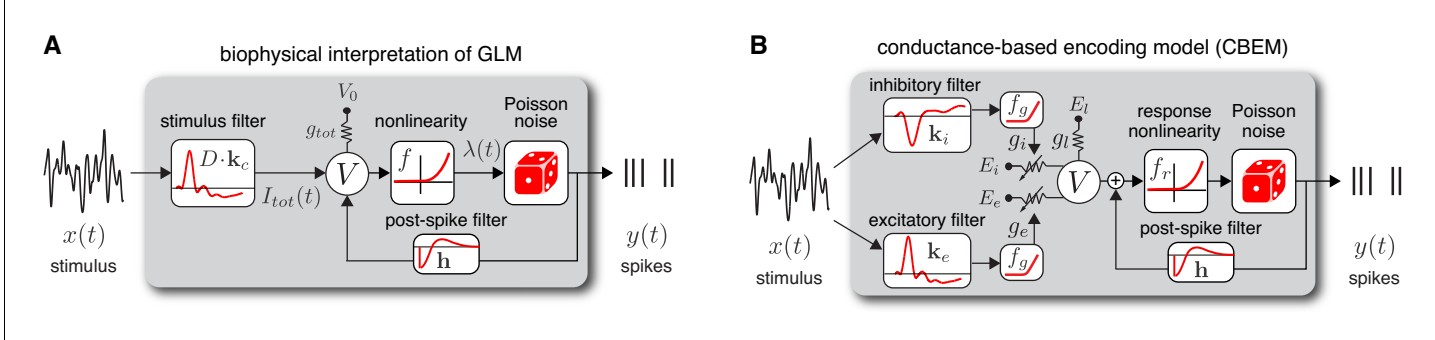

**Figure 1.** Model diagrams. (A) Diagram illustrating novel biophysical interpretation of the generalized linear model (GLM). The stimulus $x_t$ is convolved with a conductance filter **k** weighted by $D = (E_e - E_i)$, the difference between excitatory and inhibitory current reversal potentials, resulting in total synaptic current $I_{tot}(t)$. This current is injected into the linear RC circuit governing the membrane potential $V_t$, which is subject to a leak current with conductance $g_{tot}$ and reversal potential $V_0$. The instantaneous probability of spiking is governed by a the conditional intensity $\lambda_t = f(V_t)$, where $f$ is a nonlinear function with non-negative output. Spiking is conditionally Poisson with rate $\lambda_t$, and spikes gives rise to a post-spike current or filter **h** that affects the subsequent membrane potential. (B) Conductance-based encoding model (CBEM). The stimulus $\mathbf{x}_t$ is convolved with filters $\mathbf{k}_e$ and $\mathbf{k}_i$, whose outputs are transformed by rectifying nonlinearity $f_g$ to produce excitatory and inhibitory synaptic conductances $g_e(t)$ and $g_i(t)$. These time-varying conductances and the static leak conductance $g_l$ drive synaptic currents with reversal potentials $E_e$, $E_i$, and $E_l$, respectively. The resulting membrane potential $V_t$ is added to a linear spike-history term, given by $\mathbf{h} \cdot \mathbf{y}_t^{hist}$, and then transformed via rectifying nonlinearity $f_r$ to obtain the conditional intensity $\lambda_t$, which governs conditionally Poisson spiking as in the GLM. *Figure 1—figure supplement 1* shows that the CBEM parameters can be recovered from simulated data.

The online version of this article includes the following figure supplement(s) for figure 1:

**Figure supplement 1.** Convergence of model parameter fits.

spike probability as a function of membrane depolarization. Lastly, the post-spike current **h** determines how membrane potential is reset following a spike.

We can rewrite the standard GLM to emphasize this biological interpretation explicitly:

$$V_t = \mathbf{k} \cdot \mathbf{x}_t + \mathbf{h} \cdot \mathbf{y}_t^{hist} + b \quad \text{(membrane potential)} \tag{3}$$

$$\lambda_t = f_r(V_t) \quad \text{(instantaneous spike rate)} \tag{4}$$

$$y_t \mid \lambda_t \sim Poiss(\Delta\lambda_t). \quad \text{(probabilistic spiking)} \tag{5}$$

Note that in this 'soft' version of the IF model, the only noise source is the conditionally Poisson spiking mechanism; this differs from other noisy extensions of the IF model with linear current-based input and 'hard' spike thresholds, which require more elaborate methods for computing likelihoods (*Paninski, 2004*; *Pillow et al., 2005*; *Paninski et al., 2008*). To convert this model to a classic leaky integrate-and-fire model, we could replace $f_r$ with a 'hard' threshold function that jumps from zero to infinity at some threshold value of the membrane potential, set the stimulus filter **k** to an exponential decay filter, and set the post-spike filter **h** to a delta function that causes instantaneous reset of the membrane potential following a spike. The GLM membrane potential is a linear function of the input, just as in the classic leaky IF model, and thus both models fail to capture the nonlinearities apparent in the synaptic inputs to most real neurons (*Schwartz and Rieke, 2011*).

## Interpreting the GLM as a conductance-based model

Here, we propose a novel biophysically realistic interpretation of the classic Poisson GLM as a dynamical model with conductance-based input. In brief, this involves writing the GLM as a conductance-based model with excitatory and inhibitory conductances governed by affine functions of the stimulus, but constrained so that total conductance is fixed. This removes voltage-dependence of the membrane currents, making the membrane potential itself an affine function of the stimulus. The remainder of this section lays out the mathematical details of this interpretation explicitly.

Consider a neuron with membrane potential $V_t$ governed by the ordinary differential equation:

$$\frac{dV_t}{dt} = -g_l(V_t - E_l) - g_e(t)(V_t - E_e) - g_i(t)(V_t - E_i) \tag{6}$$

where $g_l$ is leak conductance, $g_e(t)$ and $g_i(t)$ are time-varying excitatory and inhibitory synaptic conductances, and $E_l$, $E_e$ and $E_i$ are the leak, excitatory and inhibitory reversal potentials.

A natural question to ask is: under what conditions, if any, is this model a GLM? Answering this question aims to reveal what biophysical assumptions the GLM implicitly enforces when modeling spike trains. Here, we provide a set of sufficient conditions for an equivalence between the two. The definition of a GLM requires the membrane potential $V_t$ to be an affine (linear plus constant) function of the stimulus, which holds if the two following conditions are met:

1. Total conductance $g_{tot}(t)$ is constant, so the membrane equation is a linear ODE.
2. The input current $I_{tot}(t)$ is an affine function of the stimulus $\mathbf{x}_t$.

The first condition implies $g_e(t) + g_i(t) = c$, for some constant $c$, and the second implies that $g_e(t)E_e + g_i(t)E_i$ is a linear function of the stimulus.

We can satisfy these two conditions simultaneously by modeling the excitatory and inhibitory conductances as affine functions of the stimulus, driven by linear filters of opposite sign:

$$
\begin{aligned}
g_e(t) &= \mathbf{k}_c \cdot \mathbf{x}_t + b_e && (\text{GLM excitatory conductance}) \\
g_i(t) &= -\mathbf{k}_c \cdot \mathbf{x}_t + b_i, && (\text{GLM inhibitory conductance})
\end{aligned}
\tag{7}
$$

where $\mathbf{k}_c$ denotes the linear 'conductance' filter, and $b_e$ and $b_i$ are arbitrary constants. Under this setting, excitatory and inhibitory conductances are driven by equal and opposite linear projections of the stimulus, with total conductance fixed at $g_{tot} = g_l + b_e + b_i$.

We can therefore rewrite the membrane equation (*Equation 6*) as:

$$
\frac{dV_t}{dt} = -g_{tot}V_t + (E_e - E_i)\mathbf{k}_c \cdot \mathbf{x}_t + b_{tot}, \qquad (\text{GLM membrane equation})
\tag{8}
$$

where $b_{tot} = b_e E_e + b_i E_i$. Setting the initial voltage to the steady-state value $V_0 = b_{tot}/g_{tot}$, the instantaneous membrane potential is then given by

$$
V_t = \mathbf{k} \cdot \mathbf{x}_t + V_0,
\tag{9}
$$

where the equivalent standard GLM filter $\mathbf{k}$ is equal to the linear convolution of $\mathbf{k}_c$ with an exponential decay filter, that is: $\mathbf{k} = \int_0^t (E_e - E_i)\mathbf{k}_c(t) e^{-g_{tot}\cdot(t-t')} dt'$. This shows that membrane potential $V_t$ is an affine function of the stimulus, so by adding a monotonic nonlinearity and conditionally Poisson spiking, the model is clearly a GLM.

Thus, to summarize, the GLM can be interpreted as a conductance-based model in which a linear filter drives equal and opposite fluctuations in excitatory and inhibitory synaptic conductances. The GLM filter $\mathbf{k}$ is equal to the convolution of this conductance filter with an exponential decay filter whose time constant is the inverse of the (constant) total conductance.

## The conductance-based encoding model (CBEM)

From this novel interpretation of the GLM, it is straightforward to formulate a more realistic conductance-based statistical spike train model. Namely, we can remove the constraint needed to construct a GLM: that excitatory and inhibitory conductance sum to a constant. Relaxing this constraint, so that total conductance can vary, results in a new model that we refer to as the *conductance-based encoding model* (CBEM). The CBEM represents an extension of GLM to allow for differential tuning of excitation and inhibition and adds rectifying nonlinearities governing the relationship between the stimulus and synaptic conductances. (See model diagram, *Figure 1b*). The CBEM model is no longer a GLM because the filtering it performs on the stimulus is nonlinear.

Formally, the CBEM is driven by excitatory and inhibitory synaptic conductances that are each linear-nonlinear functions of the stimulus:

$$
g_e(t) = f_g(\mathbf{k}_e \cdot \mathbf{x}_t + b_e) \qquad (\text{CBEM excitatory conductance})
$$

$$
g_i(t) = f_g(\mathbf{k}_i \cdot \mathbf{x}_t + b_i) \qquad (\text{CBEM inhibitory conductance}),
\tag{10}
$$

where $\mathbf{k}_e$ and $\mathbf{k}_i$ are linear filters driving excitatory and inhibitory conductance, respectively, $f_g$ is a soft-rectifying nonlinearity that ensures that conductances are non-negative (see

Materials and methods, *Equation 14*), and $b_e$ and $b_i$ determine the baseline excitatory and inhibitory conductances in the absence of input. The CBEM membrane potential $V_t$ then evolves according to the ordinary differential equation (*Equation 6*) under the influence of the two time-varying conductances $g_e(t)$ and $g_i(t)$.

To incorporate spike-history effects, we add a linear autoregressive term to the membrane potential. This results in an 'effective' membrane potential $\tilde{V}_t$ given by:

$$\tilde{V}_t = V_t + \mathbf{h} \cdot \mathbf{y}_t^{hist}, \qquad \text{(effective membrane potential)} \tag{11}$$

where $\mathbf{y}_t^{hist}$ is a vector of binned spike history at time t. We convert membrane potential to spike rate using a biophysically motivated output nonlinearity proposed by *Mensi et al. (2011)*:

$$\lambda(t) = f_r(\tilde{V}_t) = \alpha \log\left(1 + \exp\left(\frac{\tilde{V}_t - \mu}{\beta}\right)\right), \qquad \text{(output nonlinearity)} \tag{12}$$

where $\mu$ is a 'soft' spike threshold, and $\alpha$ and $\beta$ jointly determine slope and sharpness of the nonlinearity, respectively (see Materials and methods). Spiking is then a conditionally Poisson process given the rate, as in the Poisson GLM (*Equation 5*).

The CBEM is similar to the Poisson GLM in that the only source of stochasticity is the conditionally Poisson spiking mechanism: we assume no additional noise in the conductances or the voltage. This simplifying assumption, although not biophysically accurate, makes log-likelihood simple to compute, allowing for efficient maximum likelihood inference using standard ascent methods (see Materials and methods).

## Validating the CBEM modeling assumptions with intracellular data

To validate the modeling assumptions of the CBEM, we use intracellular recordings from RGCs. First, we establish that an LN model can capture the relationship between stimuli and synaptic conductances measured intracellularly (*Figure 2*). An LN model for RGC conductances is plausible because the bipolar cells that drive RGCs are known to be well-characterized by LN models (*Rieke, 2001*; *Demb et al., 2001*; *Beaudoin et al., 2008*; *Gollisch and Meister, 2010*; *Liu et al., 2017*; *Real et al., 2017*). To test the assumption in detail, we analyzed voltage clamp recordings from ON parasol RGCs in response to a full-field noise stimulus (*Trong and Rieke, 2008*). We fit the measured conductances with a linear-nonlinear model with a soft-rectified nonlinearity to account for synaptic thresholding at the bipolar-to-ganglion cell synapse (and at the amacrine cell synapses for the inhibitory inputs): $f_g(\cdot) = \log(1 + \exp(\cdot))$. The model accurately captured the relationship between projected stimuli and observed conductances on test data, accounting for 79 ± 4% (mean ± SEM) and 63 ± 3% of the variance of mean excitatory and inhibitory conductances, respectively.

Second, we establish that the output nonlinearity $f_r$, which maps membrane potential to instantaneous firing rate (*Equation 12*), provides an accurate description of the empirical relationship between membrane potential and spiking (*Figure 3*). To validate this model component, we examined dynamic current clamp recordings from two ON parasol RGCs. The dynamic clamp recordings drove RGCs with currents determined by previously measured excitatory and inhibitory conductances. To reduce noise, we computed average membrane potential over repeated presentations of the same measured conductance traces. We then computed nonparametric estimates of the nonlinearity (see Materials and methods). We found that the parametric function we assumed (*Equation 12*) closely approximated a non-parametric estimate of the nonlinearity (*Figure 3c* black; see Materials and methods for details).

Note that although previous analyses of RGC responses using Poisson GLMs have shown that an exponential nonlinearity captures the mapping from stimuli to spike rates more accurately than a rectified-linear nonlinearity (*Pillow et al., 2008*), we found the opposite here: the nonlinearity was better described with a soft-rectification function. This discrepancy may result from the fact that the GLM has a single nonlinearity, whereas the CBEM has a cascade of two nonlinearities: one mapping filter output to conductance, and a second mapping membrane potential to spike rate.

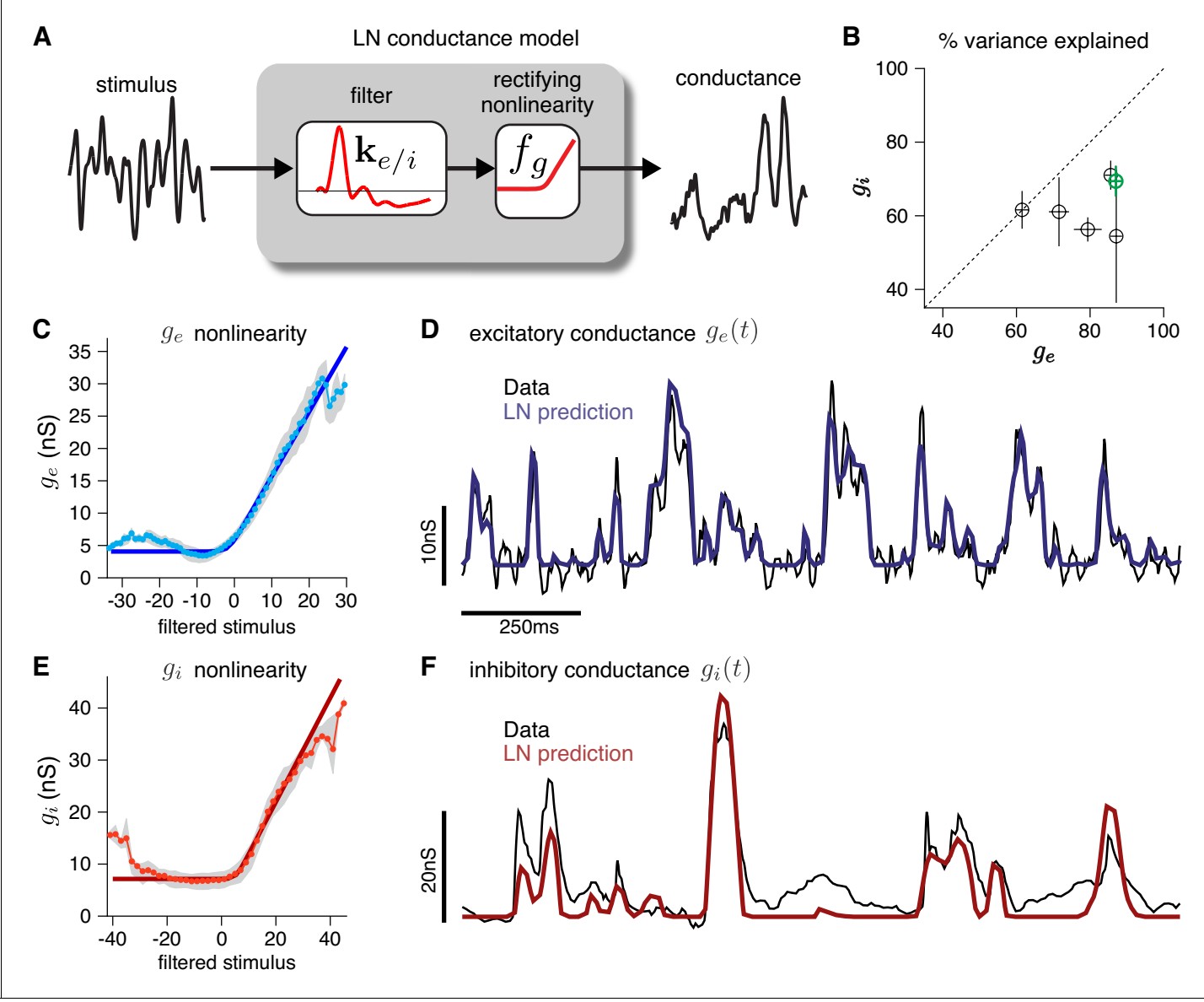

**Figure 2.** Validating the LN conductance model. The CBEM describes the relationship between stimulus and each synaptic conductance with a linear-nonlinear (LN) cascade, consisting of a linear filter followed by a fixed rectifying nonlinearity. (A) LN conductance model schematic. (B) The percent variance explained ($R^2$) for excitatory and inhibitory conductances from 6 ON parasol RGCs, computed using cross-validation with a 6 s test stimulus. Error bars indicate standard deviation across all test stimuli. (C) The excitatory conductance as a function of the filtered stimulus values for the example cell indicated in green in B. The gray region shows the middle 50-percentile of the distribution of observed excitatory conductance given the filtered stimulus value. The soft-rectifying nonlinearity (dark blue) closely matched the average conductance given the filtered stimulus value (light blue points). (D) Measured excitatory conductances in the same cell (black) and predictions from the LN model (blue) in response to a test stimulus. (E) The inhibitory conductance nonlinearity for the same neuron. The soft-rectifying nonlinearity (dark red) closely approximated the average inhibitory conductance as a function of the filtered stimulus value (light red). (F) Measured excitatory conductances (black) and the predictions of the LN model (red) on a test stimulus for the same cell.

## Predicting conductances from spikes with CBEM

We now turn to a key application of the CBEM: the inferring of excitatory and inhibitory synaptic conductances from extracellular spike train data. To test the model's ability to make such predictions, we fit the model parameters to a dataset consisting of stimuli and observed spike times. We then used the inferred filters to predict the excitatory and inhibitory conductances elicited in response to novel stimuli recorded in the same cells.

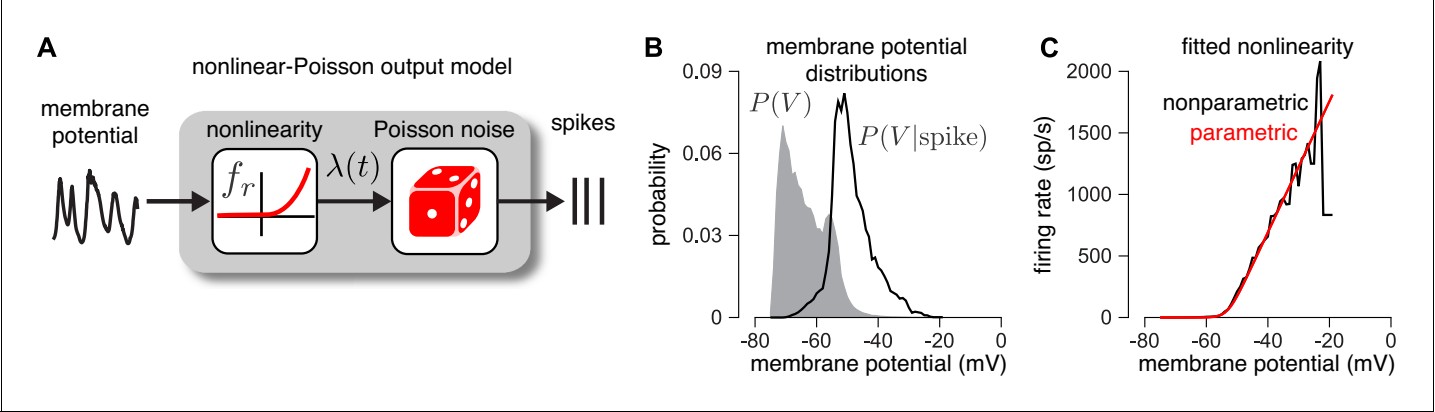

**Figure 3.** Validating the firing rate nonlinearity. (**A**) Schematic of the mapping from membrane potential to spikes under the CBEM. (**B**) The raw (gray) and spike-triggered (black) distribution of intracellular membrane potential obtained from intracellular recordings in two parasol RGCs. (**C**) Nonparametric estimate of the output nonlinearity (black trace), computed by applying Bayes' rule to the distributions in B, compared to a soft-rectified linear function (red trace).

The training data consisted of spike trains from six macaque ON-parasol RGCs obtained in cell-attached recordings with full-field white noise stimuli. Each cell was stimulated with ten unique 6 s stimulus segments, repeated three or four times each, resulting in a total of thirty to forty 6 s trials per neuron (*Trong and Rieke, 2008*). We fit the CBEM parameters (conductance filters and spike history filter) to a single cell's responses to 9 of the stimulus segments and evaluated performance using the remaining held-out segment (10-fold cross validation). Thus, the model was fit using spike trains elicited by three or four repeats of a 54 s full-field noise stimulus (see Materials and methods). For comparison, we also fit the conductance filters directly to measured excitatory and inhibitory conductances from intracellular recordings using the same stimuli and the same cross-validation procedure.

*Figure 4* shows the conductance filters estimated from intracellular data (fit to conductances) and extracellular data (fit to spike trains only) for two example cells, along with the predicted excitatory and inhibitory conductances elicited by a novel test stimulus. The filters fit to spikes were similar to those fit to conductances, and the conductance predictions from both models were highly correlated with the measured traces. *Figure 5* shows a summary statistics comparing the two models' performance for all six neurons for which we had both spike train and conductance recordings. For both models, predicted conductances traces were highly correlated with the measured conductances for all six cells. Using only a few minutes of spiking data, the conductances predicted by the extracellular model had an average correlation of r = 0.73 ± 0.01 (mean ± SEM) for the excitatory conductance and r = 0.69 ± 0.03 for the inhibitory conductance, compared to averages of r = 0.89 ± 0.02 (excitation) and r = 0.82 ± 0.01 (inhibition) for the LN model fit directly to conductances (*Figure 5a–b*).

Although the extracellular model predicted the basic timecourse of the observed conductances with high fidelity, there were small systematic discrepancies between model-predicted and measured conductances. For example, measured conductances had nearly zero lag in their cross-correlation (0.0 ± 2.4 ms; see also *Cafaro and Rieke, 2013*), whereas the predicted excitatory conductance slightly preceded the inferred inhibition for all six cells (r = 12.6 ± 1.0 ms, Student's *t*-test p < 0.0001; *Figure 5e–f*). The predicted excitation preceded the average measured excitation by 5.6 ± 0.7 ms (p = 0.0005), while the predicted inhibition showed only a slight and statistically insignificant delay compared to the measured inhibition (r = 2.5 ± 1.3 ms, p = 0.11; *Figure 5g–h*).

## Positively correlated excitation and inhibition in ON-midget cells

We also applied the CBEM to spike trains recorded from 5 ON-midget cells in response to the same type of full-field noise used for the parasol cells. In contrast to the parasol cells, ON-midget cells have positively correlated excitation and inhibition with excitation preceding inhibition (*Cafaro and Rieke, 2013*). This breaks the GLM assumption of equal and opposite tuning of the two conductances. A set of unique 6 s stimuli were used to the the model (33–35 trials for spike recordings and 5–

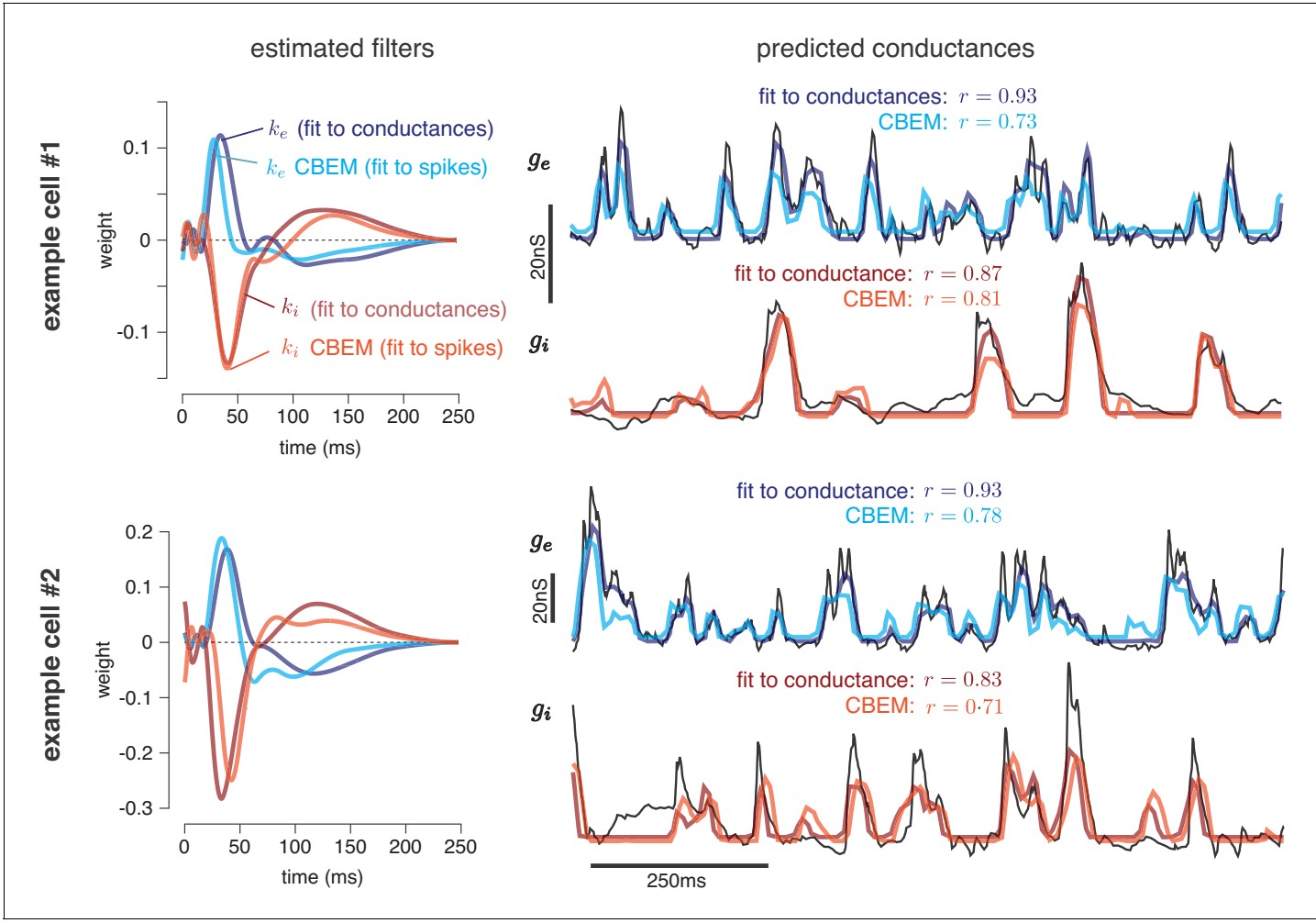

**Figure 4.** Predicting conductances from spikes with CBEM. Model parameters and conductance predictions for two example ON parasol RGCs. Left: Linear kernels for the excitatory (blue) and inhibitory (red) conductances estimated from spike train data (light red, light blue) alongside filters from an LN model fit directly to measured conductances (dark red, dark blue). The filters represent a combination of events that occur in the retinal circuitry in response to a visual stimulus, and are primarily shaped by the cone transduction process. Right: Measured conductances elicited by a test stimulus (black), along with predictions from the CBEM (fit to spikes) and LN model (fit to conductance data), indicating that the CBEM can predict synaptic conductances nearly as well as a model fit to intracellular conductance measurements. Estimated conductances and conductance filters are scaled for ease of visualization due to the presence of an unidentifiable scale factor relating to membrane capacitance. Inhibition and excitation were scaled equally.

20 trials for the LN conductance model). The models were compared to the average conductances recorded in response to a repeated novel 6 s stimulus (5–10 repeats).

The CBEM captured the tuning of the synaptic conductances received by the midget cells. An example cell is shown in *Figure 6—figure supplement 1*. The CBEM predicted the excitatory conductance with an average correlation of r = 0.85 ± 0.03 compared to the intracellular LN model with a correlation of r = 0.95 ± 0.003 (*Figure 6a*). The inhibitory conductance showed more nonlinear behavior than can be captured by a single LN unit: the CBEM predicted inhibition with r = 0.33 ± 0.06 and the LN fit to the conductance had a correlation coefficient of only r = 0.54 ± 0.07 (*Figure 6b*). The CBEM captured the fact that the inhibitory input had ON tuning, but delayed compared to excitation (*Figure 6c–d*). This was also seen in the cross-correlation between excitation and inhibition (*Figure 6e*). The data showed a cross-correlation peak with excitation preceding inhibition by 10.1 ± 0.52 ms, and the CBEM showed a similar timing difference of 8.2 ± 0.8 ms (paired Student's t-test, p = 0.06; *Figure 6f*). However, midget cells receive OFF inhibitory input in addition to

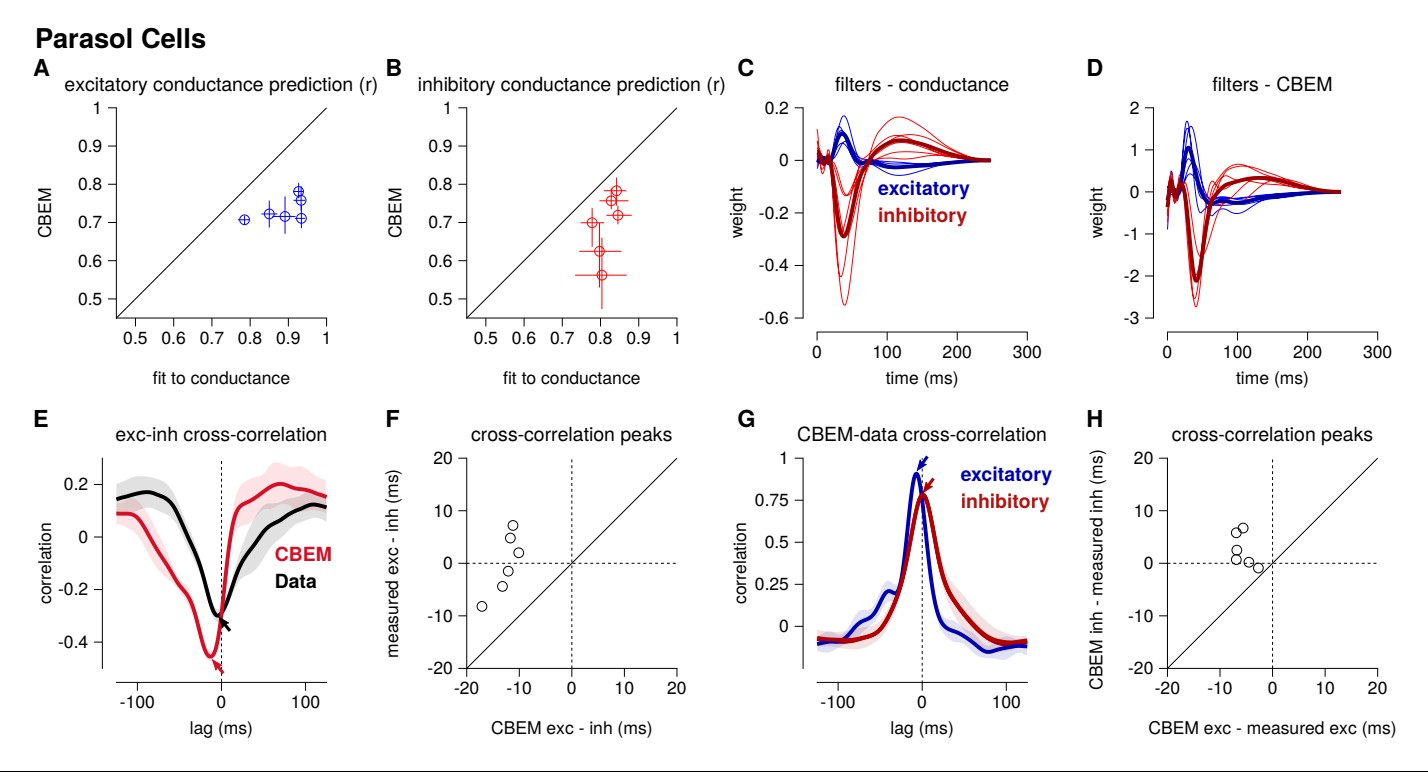

**Figure 5.** Summary of the CBEM fits to 6 ON parasol RGCs for which we had both spike train and conductance recordings. (A) The correlation coefficient (*r*) between the mean observed excitatory synaptic input to a novel 6 s stimulus and the conductance predicted by the LN cascade fit to the excitatory conductance (y-axis) compared to the CBEM prediction from spikes (x-axis) for each cell. Error bars indicate the minimum and maximum values observed across all cross-validated stimuli (B) Same as C for the inhibitory conductance. (C) The excitatory (blue) and inhibitory (red) filters estimated from voltage-clamp recordings. The thick traces show the mean filters. (D) The excitatory (blue) and inhibitory (red) filters estimated by the CBEM from spike trains. (E) The cross-correlation of the excitatory and inhibitory conductances for an example cell measured from the data (black trace; region shows standard deviation across the 10 stimuli) compared to the cross-correlation in the CBEM fit to that cell (red trace). Arrows indicate the peaks of the cross-correlations. In the data, excitation and inhibition are anti-correlated and show similar timing. However, excitation precedes inhibition in the model. (F) The cross-correlation peak times between excitation and inhibition measured from data (y-axis) compared to the conductances predicted by the CBEM (x-axis) for all 6 cells. Negative values on the x-axis indicate that excitation leads inhibition in the CBEM fits to these cells. (G) Comparing the timing of excitatory and inhibitory conductances from the data and the CBEM for the example cell in E. The cross-correlation between the measured excitatory conductance and the CBEM's excitatory conductance (blue) and the cross-correlation between data and model for the inhibitory conductances (red). (H) Cross-correlation peak times between measured and CBEM predicted inhibition (y-axis) and excitation (x-axis).

the larger ON inhibitory input (*Cafaro and Rieke, 2013*), and therefore a single LN unit could not completely capture inhibition in these cells. The true excitation was faster than the predicted excitation by r = 3.8 ± 0.8 ms (Student's t-test p = 0.008), and the measured inhibition was similarly timed with the model estimate (1.56 ± 0.6 ms, p = 0.06; *Figure 6g–h*). In summary, the CBEM can discover positive correlations between excitation and inhibition despite being initialized using a GLM with oppositely tuned excitation and inhibition (see Materials and methods).

## Characterizing spike responses with CBEM

Given the CBEM's ability to infer intracellular conductances from spike train data, we sought to examine how well it predicts spike responses to novel stimuli. Most encoding models are only tested with data from extracellular recordings, which are far easier to obtain and to sustain over longer periods. It therefore seems natural to ask: does the CBEM's increased degree of biophysical realism confer advantages for predicting spikes?

To answer this question, we fit the CBEM and classic Poisson GLM to a population of 9 extracellularly recorded macaque RGCs stimulated with full-field binary white noise (*Uzzell and Chichilnisky, 2004*; *Pillow et al., 2005*). We evaluated spike rate prediction by comparing the peri-stimulus time

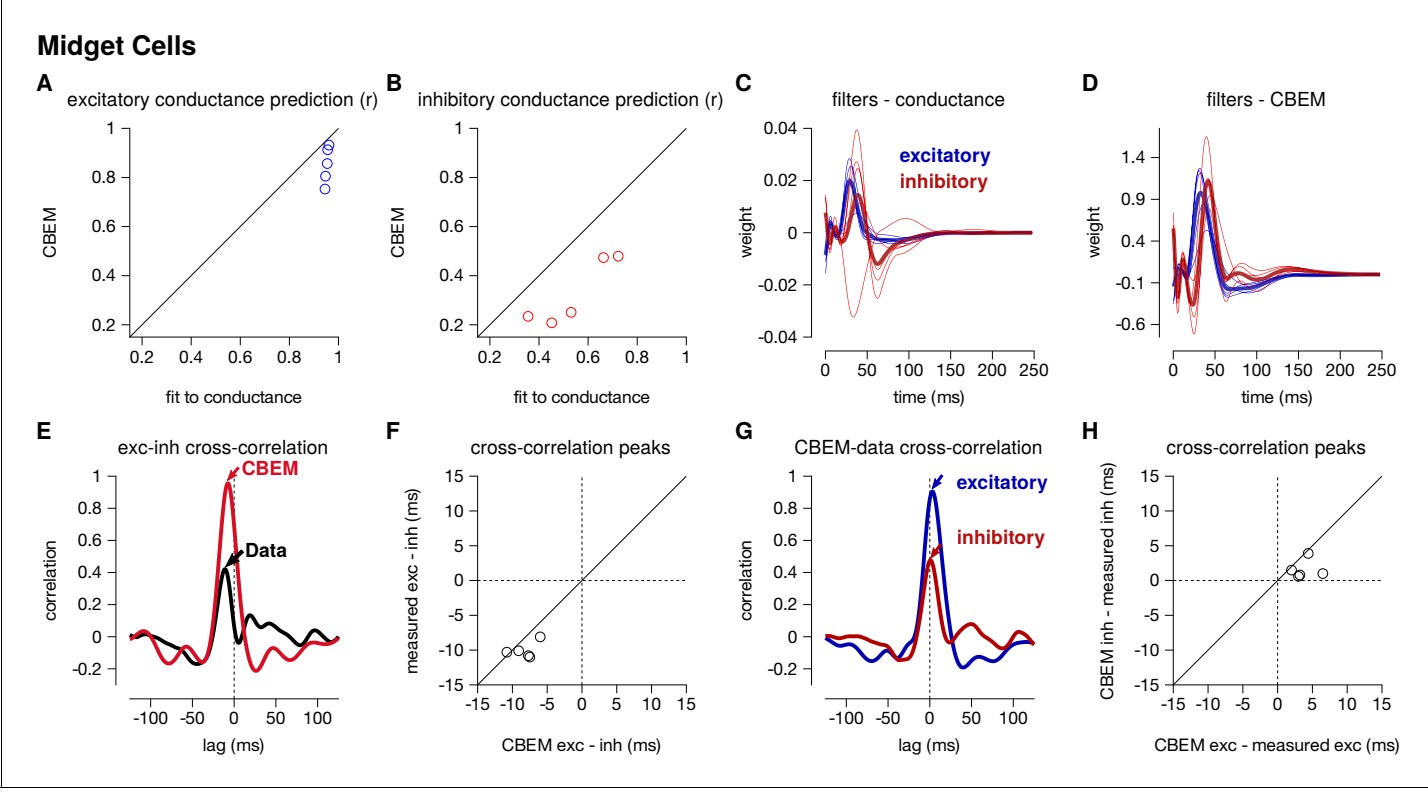

**Figure 6.** Summary of the CBEM fits to 5 ON midget RGCs. The plot follows the same conventions as the parasol results in *Figure 5*. (A,B) The correlation coefficient (*r*) between the mean observed excitatory and inhibitory synaptic input to a novel 6 s stimulus and the conductance predicted by the LN cascade fit to the excitatory conductance (y-axis) compared to the CBEM prediction from spikes (x-axis) for each cell. Conductance predictions for a single example cell are shown in *Figure 6—figure supplement 1*. The excitatory (blue) and inhibitory (red) filters estimated from voltage-clamp recordings (C) and by the CBEM from spike trains (D). (E) The cross-correlation of the excitatory and inhibitory conductances for an example cell measured from the test stimulus (data) compared to the cross-correlation predicted by the CBEM fit to that cell (red trace). (F) The cross-correlation peak times between excitation and inhibition measured from data compared to the conductances predicted by the CBEM for all five cells. (G) Comparing the timing of excitatory and inhibitory conductances from the data and the CBEM for the example cell in E. The cross-correlation between the measured excitatory conductance and the CBEM's excitatory conductance (blue) and the cross-correlation between data and model for the inhibitory conductances (red). (H) Cross-correlation peak times between measured and CBEM predicted inhibition and excitation.

The online version of this article includes the following figure supplement(s) for figure 6:

**Figure supplement 1.** CBEM fit for an example ON-midget cell with a comparison the LN models fit directly to the conductances.

histogram (PSTH) of the simulated models to the PSTH of real neurons using a 5 s test stimulus (*Figure 7*). The CBEM had higher prediction accuracy than the GLM for all nine cells, 86% of the variance of the PSTH on average vs. 77% for the GLM. We then evaluated spike train prediction by comparing log-likelihood on a 5 min test dataset. The CBEM again outperformed the GLM on all cells, offering an improvement of $0.34 \pm 0.11$ bits/spike on average over the GLM.

To gain insight into the CBEM's superior performance, we examined the average firing rate predictions of the GLM along with the average conductance predictions of the CBEM (*Figure 7c*). We found that GLM rate prediction errors (relative to the PSTH of the real neuron) were anti-correlated with the magnitude of the CBEM inhibitory conductance; the CBEM inhibitory conductance at times when the GLM spike rate exceeded the true spike rate was significantly higher than the CBEM inhibitory conductance at times when the GLM spike rate underestimated the true spike rate (t-test, $p < 0.0001$; *Figure 7—figure supplement 1b*). This suggests that the CBEM inhibitory conductance helped CBEM predictions by reducing the firing at times when the GLM over-predicted the firing rate. In contrast, the distribution of excitatory conductances did not depend on the sign of the rate prediction error (t-test, $p = 0.19$; *Figure 7—figure supplement 1a*), and the predicted excitatory conductance was positively correlated with the magnitude of the error ($r = 0.33$, $p < 0.0001$).

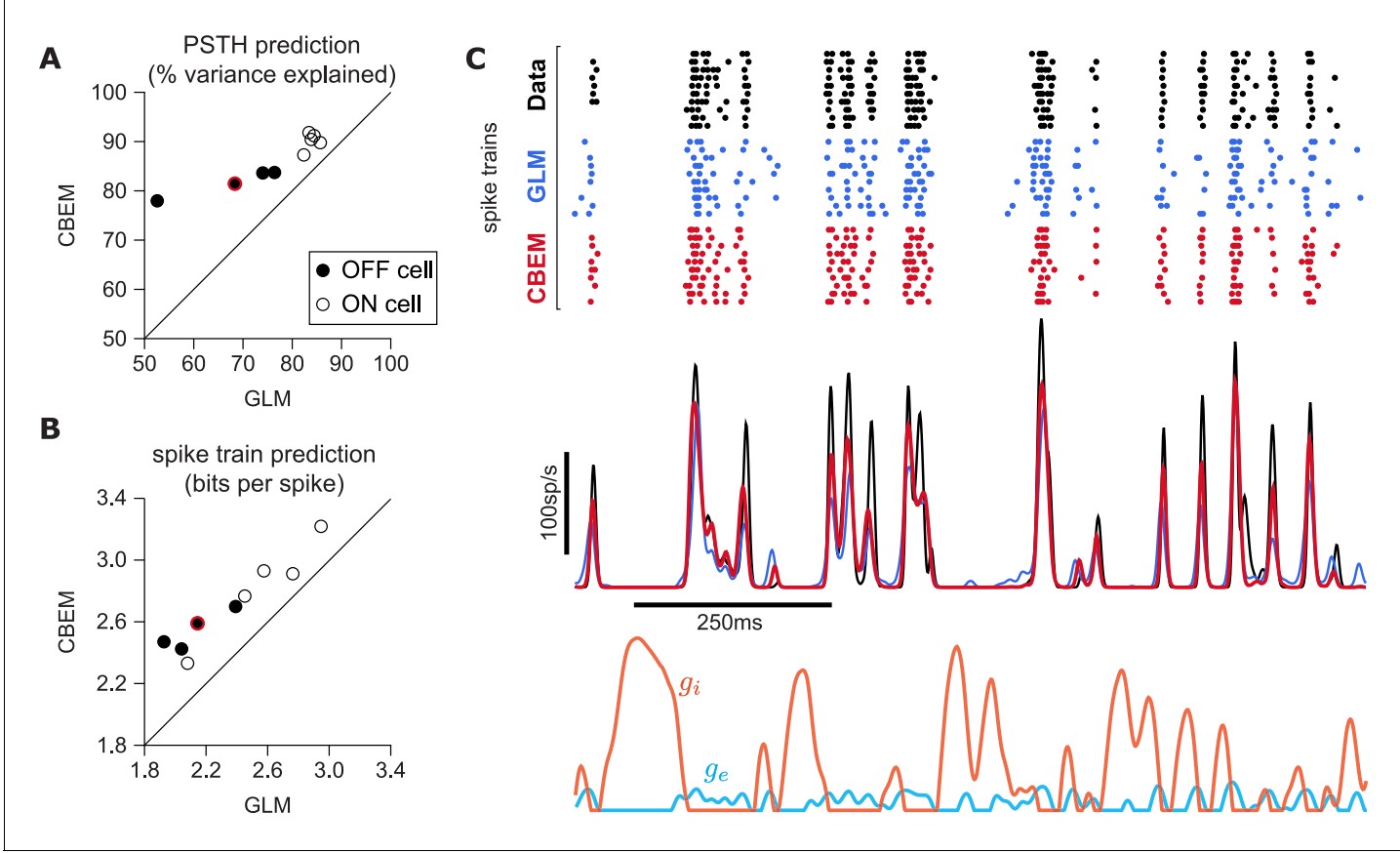

**Figure 7.** CBEM spike train predictions. (**A**) Spike rate prediction performance for the population of nine cells for 5 s test stimulus. The true rate (black) was estimated using 167 repeat trials. The red circle indicates the cell shown in C. (**B**) Log-likelihood of the CBEM compared to the GLM computed on a 5 min test stimulus. (**C**) (top) Raster of responses of an example OFF parasol RGC to repeats of a novel stimulus (black) and simulated responses from the GLM (blue) and the CBEM (red). (middle) Spike rate (PSTH) of the RGC and the GLM (blue) and CBEM (red). The PSTHs were smoothed with a Gaussian kernel with a 2 ms standard deviation. (bottom) The CBEM predicted excitatory (blue) and inhibitory (orange) conductances. The conductances are given in arbitrary units because the model does not include membrane capacitance. *Figure 7—figure supplement 1* CBEM conductance predictions.

The online version of this article includes the following figure supplement(s) for figure 7:

**Figure supplement 1.** Relation of inferred conductances to GLM prediction error.

Previous experiments have indicated that inhibition only weakly modulates parasol cell responses to full-field Gaussian noise stimuli (*Cafaro and Rieke, 2013*). To test the effect of inhibition in the model, we also refit the CBEM without any inhibitory synaptic input (CBEM$_{exc}$). We compared the excitatory filters estimated by the CBEM$_{exc}$ with the GLM filters and found that the filters are nearly identical (*Figure 8e*). This indicates that the GLM stimulus filter accounts only for the excitatory input received by the cell. The CBEM$_{exc}$ still provided a superior prediction of the PSTH than the GLM (81% of the variance explained) and an increased cross-validated log-likelihood (mean improvement of 0.14 ± 0.10 bits/sp over the GLM; *Figure 8*). The CBEM$_{exc}$ can exhibit changes in total conductance through a second, spike history independent nonlinearity (so it is not technically a GLM, as discussed in Section 3), and it predicts RGC responses better than the GLM, but not as well as the full CBEM. Thus, the full CBEM achieves superior model performance over the GLM both by including an inhibitory input, and by treating the excitatory input as a conductance-based input in a simple biophysical model.

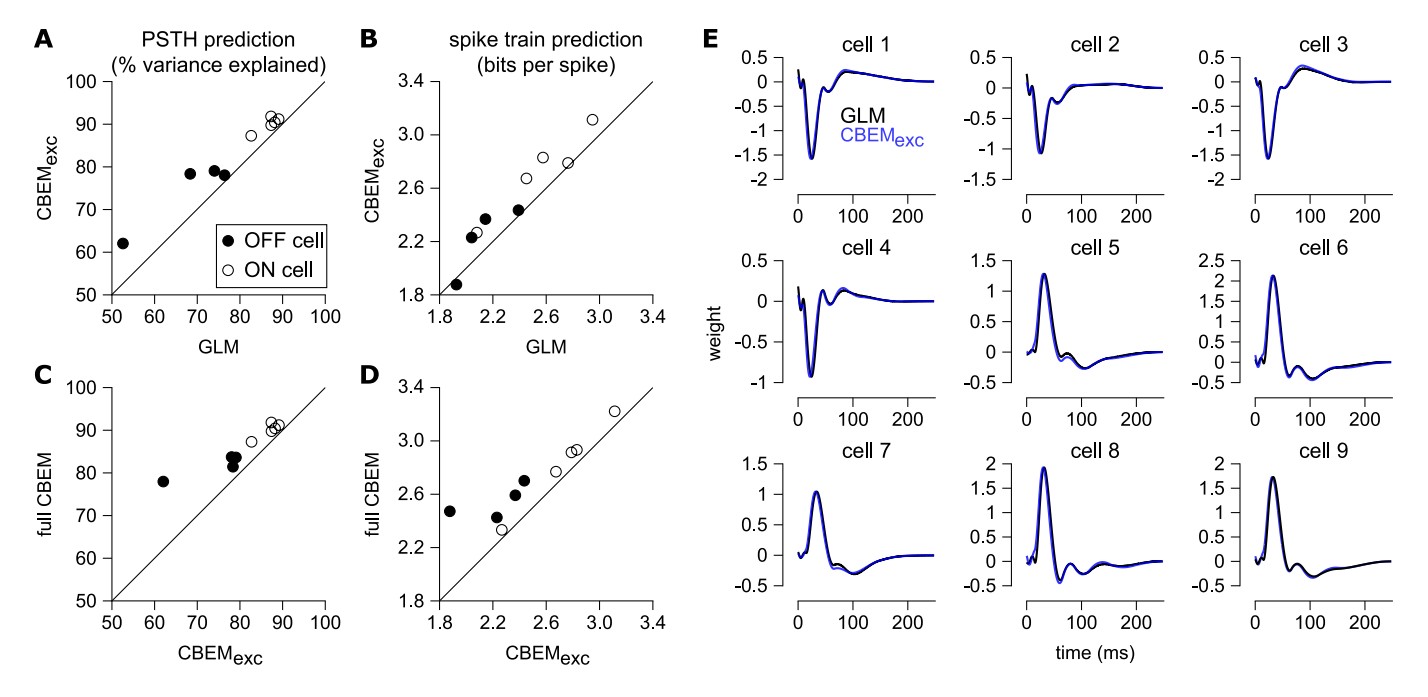

**Figure 8.** Comparison of CBEM and GLM fits. (**A**) Spike rate prediction performance and (**B**) cross-validated log-likelihood for the population of nine cells for 7 s test stimulus for the GLM and the CBEM with only an excitatory input term (CBEM$_{exc}$). (**C**) The full CBEM with inhibition shows improved spike rate predication and (**D**) cross-validated log-likelihood compared to the model without inhibition. (**E**) The GLM filters for nine parasol RGCs (black) compared to the excitatory conductance filters estimated by the CBEM without an inhibitory input (blue). The GLM filters are shown scaled to match the height of the CBEM$_{exc}$ filters.

## Capturing spike responses across contrasts

Retinal ganglion cells adapt to stimulus statistics such as contrast or variance; increases in stimulus contrast lead to decreases in gain of the neural response, allowing the dynamic range of the response to adapt to the range of contrast values present in the stimulus (*Chander and Chichilnisky, 2001*; *Fairhall et al., 2001*; *Baccus and Meister, 2002*; *Beaudoin et al., 2008*; *Mante et al., 2005*; *Garvert and Gollisch, 2013*; *Marava, 2013*; *Demb and Singer, 2015*). Understanding this phenomenon is critical for understanding how the retina codes natural stimuli, because natural scenes vary widely over contrast in both space and time. However, classic linear-nonlinear models with a single linear component fail to capture such effects. This motivates the need for a biophysically plausible modeling framework that can explain RGC responses across stimulus conditions (*Ozuysal and Baccus, 2012*; *Clark et al., 2013*; *Cui et al., 2016b*).

Previous work has shown that changes in the balance of excitatory and inhibitory input can give rise to multiplicative gain changes in neural responses (*Chance et al., 2002*; *Murphy and Miller, 2003*). This raises the possibility that the CBEM may be able account for contrast-dependent changes in RGC responses with a single set of parameters. To test this hypothesis, we fit both the CBEM and GLM to eight RGCs stimulated with full-field binary stimuli of 24%, 48%, and 96% contrast. We compared models fit simultaneously to all contrasts with models fit separately to data from each contrast. Although the CBEM does not account for many aspects of adaptation, this modeling framework allows us to test how well the LN conductance tuning alone can account for gain changes across contrasts (*Ozuysal et al., 2018*; *Latimer et al., 2019*).

To quantify the CBEM's ability to capture contrast-dependent gain changes in RGC responses, we compared GLM filters fit to RGC responses at each contrast with GLM filters fit to data simulated from the all-contrasts CBEM. (*Figure 9a*) shows GLM filters obtained at each contrast for an example RGC, while *Figure 9b* shows comparable filters fit to spikes simulated from the CBEM fit to this neuron. Both sets of filters exhibit large reductions in amplitude with increasing contrast, the key

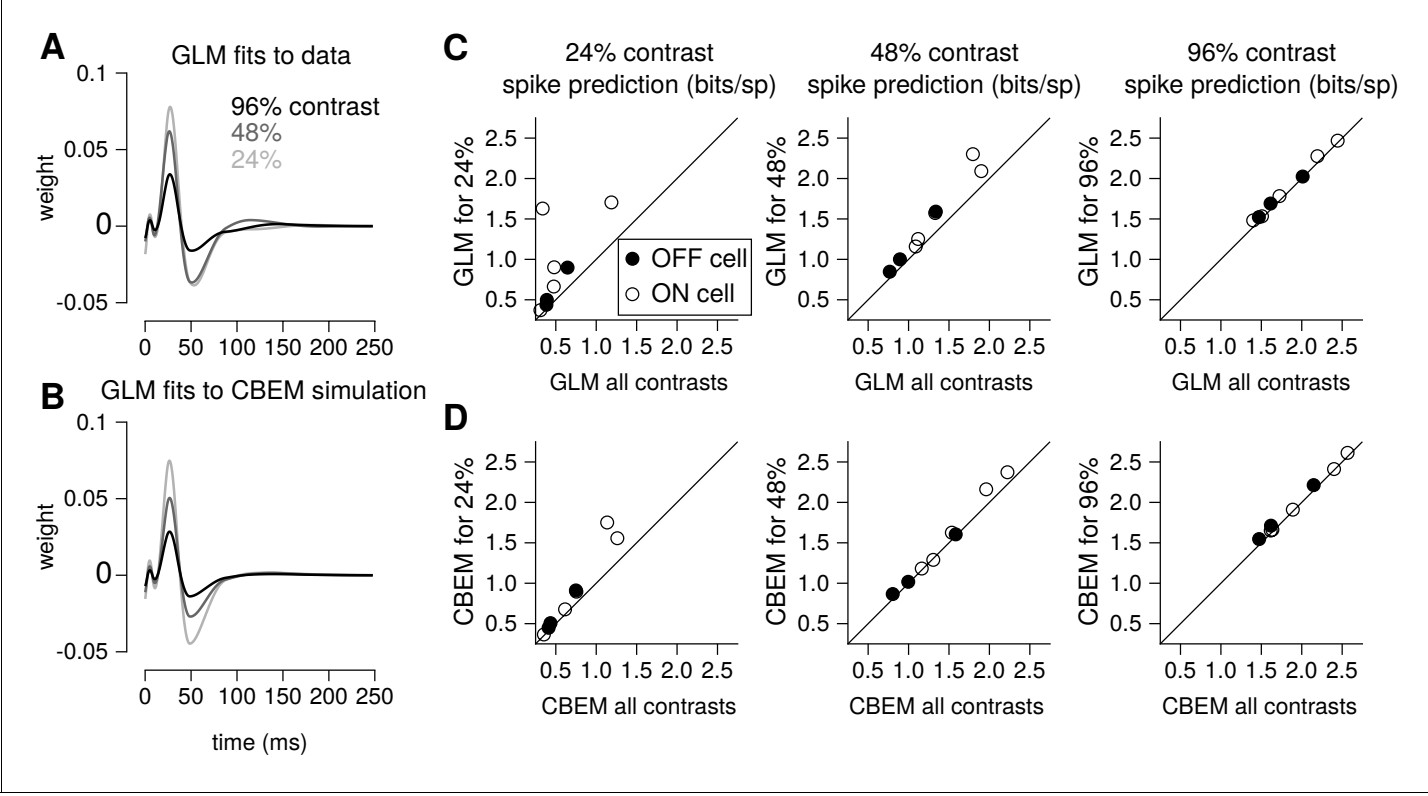

**Figure 9.** Contrast gain control in the CBEM. (A) GLM filters for an example ON cell fit to responses recorded at 24%, 48%, and 96% contrast. (B) GLM filters fit to spike trains simulated from the CBEM fit to the cell shown in A. The CBEM was fit to responses at all three contrast levels. Filter height comparisons for CBEM fits to all cells are shown in *Figure 9—figure supplement 1*. Spike train prediction performance of the (C) GLM and (D) the CBEM tested on a 4 min stimulus at 24% (left column), 48% (middle column), and 96% (right column) contrast. The model trained on all three contrast levels (y-axis) is plotted against the same class of model trained only at the probe contrast level (x-axis). *Figure 9—figure supplement 2* shows the cross-correlation between the CBEM predicted excitation and inhibition over a range of contrasts.

The online version of this article includes the following figure supplement(s) for figure 9:

**Figure supplement 1.** The filter heights (the absolute value of the peak of the filter) of the GLM fits to eight cells at all three contrast levels (one point per contrast level per cell; lines connect all contrast points from a cell), compared to the GLM filters fit the CBEM simulations of those same cells.

**Figure supplement 2.** Correlation between the CBEM's excitation and inhibition depends on contrast.

signature of contrast gain adaptation. Across all eight RGCs, we found high correlation in the filter amplitude scaling for real RGC and simulated CBEM responses ($r = 0.61$, $p < 0.05$; *Figure 9—figure supplement 1*).

We found that the CBEM maintained predictive performance across contrast levels more accurately than the GLM (*Figure 9c–d*). At 24% contrast, the GLM fit to all contrasts lost an average $0.36 \pm 0.41$ bits/sp (normalized test log-likelihood) compared to GLM fit specifically to the 24% contrast stimulus, while the CBEM lost only $0.16 \pm 0.2$ bits/sp. At 48% contrast, the GLM lost 0.20 bits/sp while CBEM only lost $0.07 \pm 0.14$ bits/sp. Finally, both models only lost $0.05 \pm 0.08$ bits/sp in the 96% contrast probe. The GLM's partial ability to generalize across these particular conditions despite having only one stimulus filter can be viewed as a consequence of our biophysical interpretation of the GLM; the GLM is equivalent to a biophysical model in which synaptic excitation and inhibition are governed by equal filters of opposite sign; *Figure 4* left shows that this assumption is approximately correct for ON parasol RGCs. However, the flexibility conferred by the slight differences in these filters with separate nonlinearities gave the CBEM greater accuracy in predicting RGC responses across a range of contrasts. We find that the correlation between excitation and inhibition in the CBEM is not constant: the CBEM predicts that the magnitude of the correlation depends on contrast *Figure 9—figure supplement 2a-b*. The CBEM predicted that, on average, excitation and inhibition were most anticorrelated at 22% contrast for the ON cells and 34% contrast for the OFF

cells *Figure 9—figure supplement 2c*. Additionally, the CBEM predicts that the mean and variance of the total synaptic conductance increases with contrast *Figure 9—figure supplement 2d-f*.

## Capturing spike responses to spatially varying stimuli

To analyze the CBEM's ability to capture responses to spatially varying stimuli, we examined a data-set of 27 parasol RGCs stimulated with spatio-temporal binary white noise stimuli (*Pillow et al., 2008*). We fit spatio-temporal filters consisting of a 5 × 5 pixel field over the same temporal extent as the models fit to full-field stimuli. The temporal profiles of excitatory and inhibitory CBEM filters were qualitatively similar to those that we observed in the full-field stimulus condition (*Figure 10a, c*). The filters were not constrained to be spatio-temporally separable (the filters were constrained to be rank 2; *Figure 10b*), which allowed the synaptic inputs to have different temporal interactions compared to the full-field stimulus.

We found that the CBEM predicted PSTHs more accurately than a Poisson GLM (83% vs. 79% average $R^2$; *Figure 10e*). The CBEM also predicted the single-trial responses with higher accuracy than the standard Poisson GLM (average improvement of $0.07 \pm 0.04$ bits/sp; *Figure 10f*). Even the CBEM with excitatory input only yielded more accurate PSTH prediction (81% $R^2$) than the GLM, but the single-trial spike train prediction fell to an average of $0.02 \pm 0.04$ bits/sp higher than the GLM (*Figure 10g–h*). Thus, the GLM predicted RGC responses to full-field noise with similar accuracy to the more complex CBEM, suggesting that the predictive performance given the training data was nearing a ceiling. Therefore, we turned to simulations to explore what type of stimuli differentiate the two models.

To gain insight into how the model's excitatory and inhibitory inputs shape the CBEM's responses to spatio-temporal stimuli, we simulated the model with uncorrelated spatio-temporal noise and with spatially correlated stimuli. The uncorrelated spatio-temporal noise was the same independent

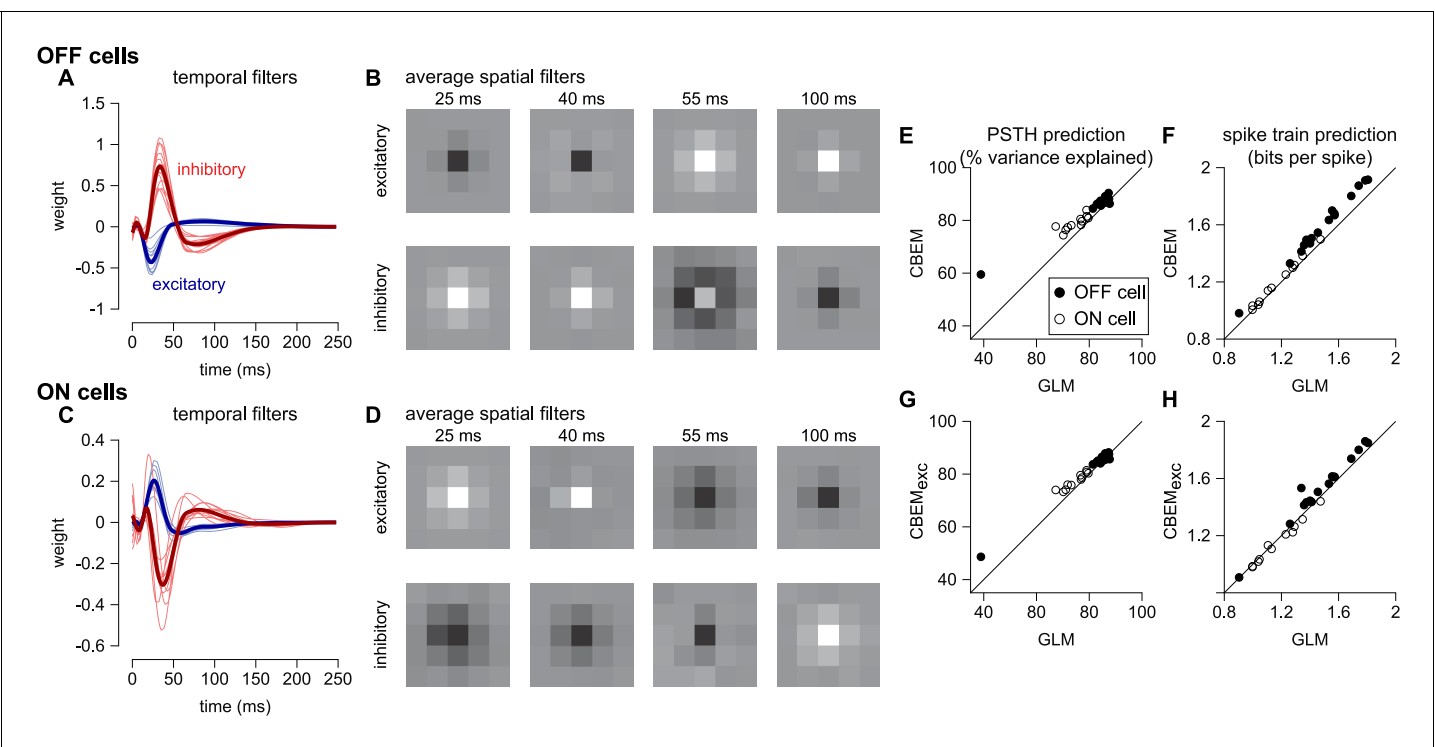

**Figure 10.** CBEM fits to a population of 27 RGCs. (**A**) Temporal profile of the excitatory (blue) and inhibitory (red) at the center pixel of the receptive field for 16 OFF parasol cells. The thick lines show the mean. (**B**) The mean spatial profiles of the excitatory (top) and inhibitory (bottom) linear filters at four different time points for the OFF parasol cells. (**C,D**) same as A,B for 11 ON parasol cells. (**E**) Spike rate prediction performance of the CBEM compared to the GLM for the population of 27 cells for 8 s test stimulus. The true rate (black) was estimated using 600 repeat trials. (**F**) Log-likelihood of the CBEM compared to the GLM computed on a 5-min test stimulus. (**G**) Spike rate prediction performance of the CBEM_exc compared to the GLM. (**H**) Log-likelihood of the CBEM_exc compared to the GLM.

binary pixel noise used in the RGC recordings, and we used a full-field and a binary center-surround stimuli for the spatially correlated noise (*Figure 11a*). Each frame of the spatially correlated center-surround stimulus was constructed by setting the center pixel to the opposite sign of the pixels in the surround, and the center pixel had equal probability of being black or white. We examined the cross-correlation of the CBEM's excitatory and inhibitory conductances in each stimulus regime and found that they were similar for the full-field and uncorrelated spatio-temporal noise stimuli (*Figure 11b* gray and black traces). In response to these two stimuli, the excitatory and inhibitory conductances showed a strong negative correlation with excitation preceding inhibition (as we saw in *Figure 5e*). The center-surround stimulus, however, produced a distinct cross-correlation pattern with a larger positive peak at the positive lags (red traces).

Finally, we simulated GLM and CBEM responses to center-surround contrast steps. The stimulus sequence started as a gray field stepping to a black center pixel with white surround for 500 ms, stepping to a gray field for 500 ms, then stepping to a white center and black surround, finally returning to a gray field (*Figure 11c* bottom). The CBEM and GLM showed similar onset responses, but the sustained responses of the CBEM simulations showed inhibition-dependent suppression for both ON and OFF cells (*Figure 11c* top and middle). The shape and sustained response of the CBE-M_exc fit to the OFF cells to center-surround steps qualitatively differed to the full CBEM: the CBE-M_exc response decayed and then rebounded slightly instead of showing only a decaying response. Thus, full-field and independent spatio-temporal noise resulted in excitatory and inhibitory

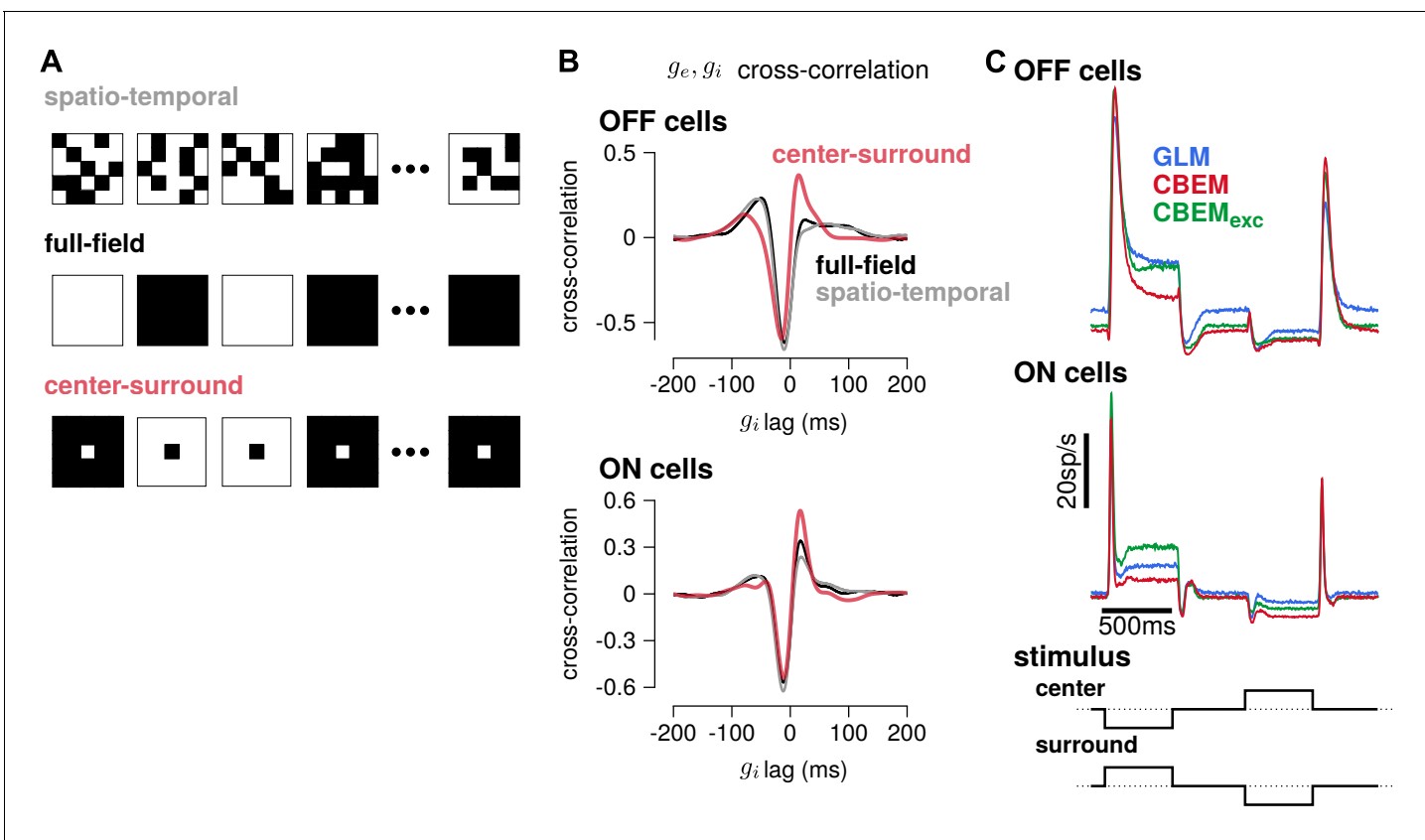

**Figure 11.** Predicted responses to spatially structured stimuli. (**A**) Example sequences of 5 × 5 pixel frames of three different types of spatiotemporal noise stimuli used to probe the CBEM. The spatio-temporal stimulus was the same binary noise stimulus used to fit the cells. The full-field stimulus consisted of binary noise at the same contrast and frame rate as the original spatio-temporal stimulus. In the opposing center-surround condition, the center pixel was of opposite contrasts to the surround pixels and the sign of the center pixel was selected randomly on each frame. (**B**) The mean cross-correlation of the CBEM predicted excitatory and inhibitory conductances for the OFF cells (top) and ON cells (bottom) in response to full-field noise (black), spatio-temporal noise (grey), and opposing center-surround noise (red). The strong negative component showed that $g_i$ is delayed and oppositely tuned compared to $g_e$. (**C**) Average firing rate of the GLM (blue), CBEM (red), and CBEM_exc (green) fits to 16 OFF cells (top) and 11 ON cells (middle) in response to opposing center-surround contrasts steps (bottom).

correlations that fit closely with the assumptions contained in the GLM. Spatial correlations, and in particular negative correlations, in the stimulus break these assumptions by co-activating excitatory and inhibitory inputs (*Cafaro and Rieke, 2013*) and therefore spatially correlated stimuli differentiate the CBEM's predictions from the GLM. These results indicate that, although the GLM captures much of the RGC responses to full-field noise, capturing the inhibitory and excitatory synaptic inputs will aid in understanding processing of naturalistic stimuli which contain spatial structure.

## Discussion

The point process GLM has found widespread use for modeling the statistical relationship between stimuli and spike trains. Here, we have offered a new biophysical interpretation of this model, showing that it can written as a conductance-based model with oppositely tuned linear excitatory and inhibitory conductances. This motivated us to introduce a more flexible and more biophysically plausible model with independent excitatory and inhibitory conductances, each given by a rectified-linear function of the sensory stimulus. This conductance-based encoding model (CBEM) is no longer technically a generalized linear model because the membrane potential is a nonlinear function of the stimulus; however, the CBEM has a well behaved point-process likelihood, making it tractable for fitting to extracellular data.

In contrast to purely statistical approaches to designing encoding models, we used intracellular measurements to motivate the choice of the nonlinear functions in the CBEM. We demonstrated that the CBEM accurately recovers the tuning of the excitatory and inhibitory synaptic inputs to RGCs purely from measured spike times. As an added bonus, it achieves improved prediction performance compared to the GLM, The interaction between excitatory and inhibitory conductances allows the CBEM to change its gain and integration time constant as a function of stimulus statistics (e.g. contrast), an effect that cannot be captured by a standard GLM. Thus, the CBEM can help reveal circuit-level computations that support perception under naturalistic conditions.

The CBEM belongs to an extended family of neural encoding models that are not technically GLMs because they do not depend on a single linear projection of the stimulus. These include multifilter LNP models with quadratic terms (*Schwartz et al., 2002*; *Rust et al., 2005*; *Park and Pillow, 2011*; *Fitzgerald et al., 2011*; *Park et al., 2013*; *Rajan et al., 2013*) or general nonparametric nonlinearities (*Sharpee et al., 2004*; *Williamson et al., 2015*); models with input nonlinearities (*Ahrens et al., 2008*) and multilinear context effects (*Williamson et al., 2016*); models inspired by deep learning methods (*McIntosh et al., 2016*; *Maheswaranathan et al., 2018*); and models with biophysically inspired forms of nonlinear response modulation (*Butts et al., 2011*; *Ozuysal and Baccus, 2012*; *McFarland et al., 2013*; *Cui et al., 2016b*; *Real et al., 2017*). The CBEM has most in common with this last group of models. Although more flexible LNLN models may predict spike trains with higher accuracy, the CBEM stands as the only model so far to have directly linked model components fit to spikes alone to experimentally measured conductances.

Although the CBEM represents a step toward biophysical realism, it still lacks many properties of real neurons. For instance, the CBEM's linear-rectified conductance does not capture the non-monotonic portions of the stimulus-conductance nonlinearities observed in the data (*Figure 2c,e*); this non-monotonicity likely arises from the fact that amacrine cells can receive inputs from both ON and OFF channels (*Manookin et al., 2008*; *Cafaro and Rieke, 2013*). Further developments to the CBEM can include additional sets of nonlinear inputs (*McFarland et al., 2013*; *Maheswaranathan et al., 2018*; *Real et al., 2017*). Such extensions could include multiple spatially distinct inputs to account for input from different bipolar cells (*Schwartz et al., 2012*; *Freeman et al., 2015*; *Vintch et al., 2015*; *Turner and Rieke, 2016*; *Liu et al., 2017*), and spatially selective rectification of inhibitory inputs that helps determine RGC responses to spatial stimuli (*Brown and Masland, 2001*; *Cafaro and Rieke, 2013*; *Schwartz and Rieke, 2013*). The model could also be extended to study pre-synaptic inhibition of the excitatory conductance, which can shape the spatial tuning of excitation (*Turner et al., 2018*) and contrast adaptation (*Cui et al., 2016b*). Adaptation can occur in localized regions of a ganglion cell's RF (*Garvert and Gollisch, 2013*), suggesting that the linear-nonlinear synaptic input functions in the CBEM should be allowed to vary over time. Additionally, future work could apply the CBEM to study the role of active conductances that depend spike history, such as an after hyper-polarization current (*Johnston et al., 1995*; *Badel et al., 2008*; *Lundstrom et al., 2008*), and recent work has shown that the parameters of

Hodgkin-Huxley style biophysical models can in some instances be recovered from spike trains alone (*Meng et al., 2011*). Spike-dependent conductances could also be examined in multi-neuron recordings; although the analyses presented here focused on the coding properties of single neurons, many of the RGCs analyzed were recorded simultaneously (*Pillow et al., 2008*; *Volgushev et al., 2015*).

Another aspect of the CBEM that departs from biophysical realism is that all stochasticity is confined to the spike generation mechanism. The CBEM models conductances and membrane potential as deterministic functions of the stimulus, which makes the likelihood tractable and allows for efficient fitting with standard conjugate-gradient methods (*Real et al., 2017*). However, the reliability of RGC spike trains depends on the stochasticity of synaptic conductances (*Murphy and Rieke, 2006*), and noise correlations between excitatory and inhibitory conductances may also affect encoding in RGCs (*Cafaro and Rieke, 2010*). A latent variable approach could be used to to incorporate stochasticity in conductances and membrane potential (*Meng et al., 2011*; *Paninski et al., 2012*; *Lankarany et al., 2016*).

We expect that the CBEM may also be useful for regions beyond the retina. Previous work on the prediction of membrane potential in primary visual cortex suggests that the CBEM could apply to neurons further along in the visual stream (*Mohanty et al., 2012*; *Tan et al., 2011*). The model could also be applied to non-visual areas such as primary auditory cortex, where different tuning motifs of excitation and inhibition are of interest (*Scholl et al., 2010*). Future work might extend the CBEM to use deeper, nonlinear cascade models to predict conductances, as opposed to the simple LN cascade we have assumed here. For example, one might use the LNLN models of the lateral geniculate nucleus (e.g. *Butts et al., 2011*; *McFarland et al., 2013*) as providing the drive to synaptic conductances in V1 neurons. This principle can extend to higher sensory regions, such as the middle temporal cortex where cascade models can approximate the inputs received from V1 (*Rust et al., 2006*). Applications in cortex may also incorporate additional inputs to the model such as local field potential, which is thought to reflect the total synaptic drive to a region (*Einevoll et al., 2013*; *Haider et al., 2016*; *Cui et al., 2016a*).

Applications of the CBEM to new brain areas could involve testing the accuracy of conductance predictions with a small number of intracellular recordings, and then applying the model to larger set of extracellular recordings with a wider range of stimuli. Although the model's simplifying assumptions limit the ability to make strong conclusions about the conductances estimated from spikes alone, the model may nevertheless guide experimental design and theories of sensory processing when intracellular recordings are unavailable.

Future work will require modeling the neural code using naturalistic stimuli, where the GLM has been shown to fail (*Carandini et al., 2005*; *van Hateren et al., 2002*; *Butts et al., 2007*; *Heitman et al., 2016*; *Turner and Rieke, 2016*). Modeling tools must also provide a link between the neural code and computations performed by the neural circuit. As we move toward stimuli with complex spatio-temporal statistics, the ability to connect distinct synaptic conductances to spiking will provide an essential tool for deciphering the complex, nonlinear neural code in sensory systems.

## Materials and methods

### Electrophysiology

We analyzed four sets of parasol RGCs. All data were obtained from isolated, peripheral macaque monkey, *Macaca mulatta*, retina.

### Synaptic current recordings

We analyzed the responses of 6 ON parasol cells previously described in *Trong and Rieke (2008)*. Cell-attached and voltage clamp recordings were performed to measure spike trains and excitatory and inhibitory currents in the same cells. The stimulus, delivered with an LED, consisted of a one dimensional, full-field white noise signal, filtered with a low pass filter with a 60 Hz cutoff frequency, and sampled at a 0.1ms resolution. Spike trains were recorded using 10 unique 6 s stimuli, and each stimulus was repeated three or four times. After the spike trains were recorded, the excitatory and inhibitory synaptic currents to the same stimuli were measured using voltage clamp recordings. Active conductances intrinsic to the RGC were blocked during these recordings and the holding

potential was set to isolate either the excitatory or inhibitory inputs received by the cell. For four of the cells, two to four trials were recorded for each of the 10 stimuli for the excitatory and inhibitory currents. For the two remaining cells, three to four excitatory current trials were recorded for all 10 stimuli and one to two trials for the inhibitory current were obtained for 8 of the stimuli. Conductances were estimated by dividing the current by the approximate driving force (−70 mV for the excitatory currents, and 70 mV for the inhibitory).

The 5 ON-midget cells were recorded as described previously (*Dunn et al., 2006*; *Trong and Rieke, 2008*; *Cafaro and Rieke, 2013*). Retinas were obtained through the Tissue Distribution Program of the Regional Primate Research Center at the University of Washington and procedures were approved by the Institutional Animal Care and Use Committee. The same type of full-field noise stimuli were used for the midget cells as with the parasol cells, and the recordings were again divided into 6 s trials. Spike trains were obtained with cell attached recordings. For each cell, 33–35 trials of unique stimuli were recorded, and 10 (for 4 cells) or 20 (for 1 cell) trials were recorded in response to a repeated stimulus. Excitatory and inhibitory currents were recorded for 5–20 trials each for non-repeated stimuli, and 5–10 trials were recorded in response to the repeated validation stimulus. Conductances were again estimated by dividing the current by the approximate driving force (−70 mV for the excitatory currents, and 70 mV for the inhibitory).

## Dynamic clamp recordings

The membrane potentials of 2 ON parasol retinal ganglion cells were recorded during dynamic clamp experiments previously reported in *Cafaro and Rieke (2013)*. The cells were current clamped and current was injected into the cells according to the equation

$$I(t) = g_e(t)(V(t - \Delta_t) - E_e) + g_i(t)(V(t - \Delta_t) - E_i) \tag{13}$$

where $g_e$ and $g_i$ were conductances recorded in RGCs in response to a light stimulus. The injected current at time $t$ was computed using the previous measured voltage with offset $\Delta t$ = 100 μs. The reversal potentials were $E_e = 0$ mV and $E_i = -90$ mV.

For the first cell, 18 repeat trials were recorded for a 19 s stimulation, and 24 repeat trials were obtained from the second cell.

## RGC population recordings: full-field stimulus

We analyzed data from two experiments previously reported in *Uzzell and Chichilnisky (2004)* and *Pillow et al. (2005)*. The first experiment included nine simultaneously recorded parasol RGCs (5 ON and 4 OFF). The stimulus consisted of a full-field binary noise stimulus (independent black and white frames) with a root-mean-square contrast of 96%. The stimulus was displayed on a CRT monitor at a 120 Hz refresh rate and the contrast of each frame was drawn independently. A 10 min stimulus was obtained for characterizing the cell responses, and a 5-min segment was used to obtain a cross-validated log-likelihood. Spike rates were obtained by recording 167 repeats of a 7.5 s stimulus.

In a second experiment, eight cells (3 ON and 5 OFF parasol) were recorded in response to a full-field binary noise stimulus (120 Hz) at 24%, 48%, and 96% contrast. An 8 min stimulus segment at each contrast level was used for model fitting, and cross-validated log-likelihoods were obtained using a novel 4 min segment at each contrast level.

## RGC population recordings: spatio-temporal stimulus

We analyzed 11 ON and 16 OFF parasol RGCs which were previously reported in *Pillow et al. (2005)*. The stimulus consisted of a spatio-temporal binary white noise pattern (i.e. a field of independent white and black pixels). The stimulus was 10 pixels by 10 pixels (pixel size of 120 μm × 120 μm on the retina), and the contrasts of each pixel was drawn independently on each frame (120 Hz refresh rate). The root-mean-square contrast of the stimulus was 96%.

A 10-min stimulus was obtained for characterizing the cell responses, and a 5-min segment was used to obtain a cross-validated log-likelihood. Spike rates were obtained by recording 600 repeats of a 10 s stimulus.

## Modeling methods

### The conductance-based encoding model

The CBEM introduced above models the spike train response of a RGC to a visual stimulus as a Poisson process where the spike rate is a function of the membrane potential (*Figure 1b*). The membrane potential is approximated by considering a single-compartment neuron with linear membrane dynamics and conductance-based input (*Equation 6*). Note that we have ignored capacitance, which would provide an (unobserved) scaling factor on $dV/dt$, but will not affect our results. The synaptic inputs (*Equation 10*) take the form of linear-nonlinear functions of the stimulus, $\mathbf{x}$, where $f_g$ is a nonlinear function ensuring positivity of the conductances. We will assume a 'soft-rectification' nonlinearity given by

$$f_g(z) = \log(1 + \exp(z)),\tag{14}$$

which behaves like a smooth version of a linear half-rectification function.

Given the conductances, we could then obtained the membrane voltage. We use a first-order exponential integrator method to solve this equation, which is exact under the assumption that $g_e(t)$ and $g_i(t)$

$$V(t + \Delta) = \exp(-\Delta g_{tot}(t))\left(V_t - \frac{I_{tot}(t)}{g_{tot}(t)}\right) + \frac{I_{tot}(t)}{g_{tot}(t)},\tag{15}$$

where

$$g_{tot}(t) = g_e(t) + g_i(t) + g_l\tag{16}$$

$$I_{tot}(t) = g_e(t)E_e + g_i(t)E_i + g_lE_l,\tag{17}$$

for $g_{tot}(t)$ and $I_{tot}(t)$, and assuming $V(0) = E_l$ at the start of each experiment.

For a set of spike times $s_{1:n_{sp}}$ in the interval $[0, S]$ and parameters $\Theta$, the log-likelihood in continuous time is

$$\log p(s_{1:n_{sp}} \mid \mathbf{x}_{[0,S]}, \Theta) = \sum_{i=1}^{n_{sp}} \log(\lambda(s_i)) - \int_0^S \lambda(t)dt\tag{18}$$

where the spike rate, $\lambda(t)$, is a function of the voltage plus spike history (*Equation 12*). This likelihood can be discretely approximated as the product of $T$ Bernoulli trials in bins of width $\Delta$ (*Citi et al., 2014*)

$$\log p(y_{1:T} \mid \mathbf{x}_{1:T}, \Theta) = \sum_{t=1}^{T} y_t \log(1 - \exp(-\lambda_t \Delta)) - (1 - y_t)\lambda_t \Delta\tag{19}$$

where $y_i = 1$ if a spike occurred in the $i$th bin and 0 otherwise.

The membrane voltage (and firing rate) is computed by integrating the membrane dynamics equation (*Equation 6*). In practice, we evaluate $V$ along the same discrete lattice of points of width $\Delta$ ($t = 1, 2, 3, \ldots T$) that we use to discretize the log-likelihood function. Assuming $g_e$ and $g_i$ remain constant within each bin, the voltage equation becomes a simple linear differential equation which we solve according to *Equation 15*.

The model parameters we fit were $\mathbf{k}_e$, $\mathbf{k}_i$, $b_e$, $b_i$, and $\mathbf{h}$, which were selected using conjugate-gradient methods to maximize the log-likelihood.

The reversal potential and leak conductance parameters were kept fixed at $E_e = 0mV$, $g_l = 200$, $E_l = -60mV$, and $E_i = -80mV$. For modeling the cells in which we had access to intracellular recordings, we set the time bin width to $\Delta = 0.1ms$ to match the sampling frequency of the synaptic current recordings. For the remaining cells, which were recorded in separate experiments, we set $\Delta = 0.083ms$, 100 times the frame rate of the visual stimulus.

The stimulus filters spanned over 100 ms, or over 1000 time bins. Therefore, we restricted the excitation and inhibitory filters to a low dimensional basis to limit the total number of free

parameters in the model. The basis consisted of 10 raised cosine 'bumps' (*Pillow et al., 2005*; *Pillow et al., 2008*) of the form

$$\mathbf{b}_j(t) = \begin{cases} \frac{1}{2}\cos\left(\frac{\log[t+c]-\phi_j}{a}\right) + \frac{1}{2} & \text{for } \frac{\log[t+c]-\phi_j}{a} \in [-\pi, \pi] \\ 0 & \text{otherwise} \end{cases} \tag{20}$$

where $t$ is in seconds. We set c = 0.02 and $a = 2(\phi_2 - \phi_1)/\pi$. The $\phi_j$ were evenly spaced from $\phi_1 = \log(0.0 + c)$, $\phi_{10} = \log(0.150 + c)$ so that the peaks of the filters spanned 0 ms to 150 ms. The spike history filter was also represented in a low-dimensional basis. The refractory period was accounted for with five square basis functions of width 0.4 ms, spanning the period $0 - 2$ ms after a spike. The remaining spike history filter consisted of 7 raised cosine basis functions (c = 0.0001) with filter peaks spaced from 2 ms to 90 ms.

The log-likelihood function for this model is not concave in the model parameters, which increases the importance of selecting a good initialization point compared to the GLM. We initialized the parameters by fitting a simplified model which had only one conductance with a linear stimulus dependence, $g_{lin}(t) = \mathbf{k}_{lin}^\top \mathbf{x}_t$ (note that this allowed for negative conductance values). We initialized this filter at 0, and then numerically maximized the log-likelihood for $\mathbf{k}_{lin}$. We then initialized the parameters for the complete model using $\mathbf{k}_e = c\mathbf{k}_{lin}$ and $\mathbf{k}_i = -\mathbf{k}_{lin}$, thereby exploiting a mapping between the GLM and the CBEM (see Results).

When fitting the model to real spike trains, one conductance filter would occasionally become dominant early in the optimization process. This was likely due to the limited amount of data available for fitting, especially for the cells that were recorded intracellularly. The intracellular recordings clearly indicated that the cells received similarly scaled excitatory and inhibitory inputs. To alleviate this problem, we added a penalty term, $\phi$, to the log-likelihood to the $L_2$ norms of $\mathbf{k}_e$ and $\mathbf{k}_i$:

$$\phi(\mathbf{k}_e, \mathbf{k}_i) = \frac{1}{2}\left(\lambda_e||\mathbf{k}_e||^2 + \lambda_i||\mathbf{k}_i||^2\right) \tag{21}$$

Thus, we maximized

$$\mathcal{L}(\theta) = \log p(y_{1:T}|\mathbf{x}_{1:T}, \mathbf{k}_e, \mathbf{k}_i, b_e, b_i) - \phi(\mathbf{k}_e, \mathbf{k}_i) \tag{22}$$

All cells were fit using the same penalty weights: $\lambda_e = 1$ and $\lambda_i = 0.2$. We note that unlike the typical situation with cascade models that contain multiple filters, intracellular recordings can directly measure synaptic currents. Future work with this model could include more informative, data-driven priors on $\mathbf{k}_e$ and $\mathbf{k}_i$.

In several analyses, we fit the CBEM without the inhibitory conductance, labeled as the CBEM$_{exc}$. All the fixed parameters used in the full CBEM were held at the same values in the CBEM$_{exc}$.

Code for fitting the CBEM has been made available at https://github.com/pillowlab/CBEM (*Latimer, 2018*; copy archived at https://github.com/elifesciences-publications/CBEM).

## Fitting the CBEM to simulated spike trains

To examine the performance of our numerical maximum likelihood estimation of the CBEM, we fit the parameters to simulated spike trains from the model with known parameters (*Figure 1—figure supplement 1*). Our first simulated cell qualitatively mimicked experimental RGC datasets, with input filters selected to reproduce the stimulus tuning of macaque ON parasol RGCs (excitation oppositely tuned and delayed compared to excitation, or 'crossover' inhibition). The second simulated cell had similar excitatory tuning, but the inhibitory input had the same tuning as excitation with a short delay. The stimulus consisted of a one dimensional white noise signal, binned at a 0.1 ms resolution, and filtered with a low-pass filter with a 60 Hz cutoff frequency. We validated our maximum likelihood fitting procedure by examining error in the fitted filters, and evaluating the log-likelihood on a 5-min test set. With increasing amounts of training data, the parameter estimates converged to the true parameters for both simulated cells. Therefore, standard fast and non-global optimization algorithms can reliably fit the CBEM to spiking data, despite the fact that the model does not have the concavity guarantees of the standard GLM.

## Fitting the conductance nonlinearity

We selected the nonlinear function $f_g$ governing the synaptic conductances by fitting a linear-nonlinear cascade model to intracellularly measured conductances evoked during visual stimulation (*Hunter and Korenberg, 1986*; *Paninski et al., 2012*; *Park et al., 2013*; *Barreiro et al., 2014*). We modeled the mean conductance $\bar{g}_e(t)$ as

$$\bar{g}_e(t) = a_e f_g((\mathbf{k}_e * \mathbf{x})(t) + b_e) + \epsilon_t \tag{23}$$

$$\epsilon_t \sim \mathcal{N}(0, \sigma^2) \tag{24}$$

where $\mathbf{x}$ is a full-field temporal stimulus, and $a_e$ and $b_e$ are constants. We selected a fixed function for the nonlinearity $f_g$. Thus, we chose the $\mathbf{k}_e$, $a_e$, and $b_e$ that minimized the squared error between the LN prediction and the measured excitatory conductance.

The soft-rectifying function was selected to model the conductance nonlinearity;

$$f_g(s) = \log(1 + \exp(s)). \tag{25}$$

We chose to fix these nonlinearities to known functions rather than fitting with a more flexible empirical form (e.g., *Ahrens et al., 2008*; *McFarland et al., 2013*). Fixing these nonlinearities to a simple, closed-form function allowed for fast and robust maximum likelihood parameter estimates while still providing an excellent description of the data.

## Fitting the spike-rate nonlinearity

We used a spike-triggered analysis (*de Boer and Kuyper, 1968*) on membrane voltage recordings to determine the spike rate nonlinearity, $f_r$, as a function of voltage for the CBEM. This is the same procedure for estimating the LN-model nonlinearity proposed in *Chichilnisky (2001)*; *Mease et al. (2013)*, but substituting the filtered stimulus with the average voltage measured across trials. The membrane potential and spikes were recorded in dynamic-clamp experiments over several repeats of simulated conductances for two cells. We computed the mean voltage recorded over all runs of the dynamic-clamp condition, which largely eliminated the action potential shapes from the voltage trace. Using the spike times from all the repeats, we computed the probability of a spike occurring in one time bin given the mean voltage, $\bar{V}$:

$$p(Sp|\bar{V}) = \frac{p(\bar{V}|Sp)p(Sp)}{p(\bar{V})} \tag{26}$$

where $p(\bar{V}|Sp)$ is the spike-triggered distribution of the membrane potential. The distribution over voltage in all times bins is $p(\bar{V})$. The spike rate (in terms of spikes per bin) is $p(Sp)$. We combined the spike times and voltage distributions for the two cells to compute a common spike rate function.

We then obtained a least-squares fit to approximate the nonlinearity with a soft-rectification function of the the form

$$p(Sp|\bar{V}) \approx f_r(t)\Delta \tag{27}$$

$$f_r(t) = \alpha \log\left(1 + \exp\left(\frac{(V_t - \mu)}{\beta}\right)\right). \tag{28}$$

The parameters fit to the empirical spike-rate nonlinearity was $\alpha = 90\,sp/s$, $\mu = -53\,mV$ and $\beta = 1.67\,mV$.

We chose to fit the spike-rate nonlinearity with the average voltage recorded over repeat data, instead of looking at the voltage in bins preceding spikes (*Jolivet et al., 2006*). The average voltage is closer in spirit to the voltage in our model than the single-trial voltage, because the voltage dynamics we considered (*Equation 6*) did not include noise nor post-spike currents.

## Generalized linear models

For a baseline comparison to the CBEM, we also fit spike trains with a GLM. We used the same Bernoulli discretization of the point-process log-likelihood function for the GLM as we did with the CBEM:

$$\log p(y_{1:T}|\mathbf{x}_{1:T}, \mathbf{k}, b, \mathbf{h}) = \sum_{t=1}^{T} y_t \log(1 - \exp(-\lambda_t \Delta)) - (1 - y_t)\lambda_t \Delta \tag{29}$$

where the firing rate is

$$\lambda_t = f_r((\mathbf{k} * \mathbf{x})(t) + b + (\mathbf{h} * \mathbf{y}^{hist})(t)). \tag{30}$$

The stimulus filter is $\mathbf{k}$ and the spike history filter is $\mathbf{y}^{hist}$. We used conjugate-gradient methods to find the maximum likelihood estimates for the parameters. We set $f_r(\cdot) = \exp(\cdot)$, which is the canonical inverse-link function for Poisson GLMs. We confirmed previous results that the soft-rectifying nonlinearity, $f_r(\cdot) = \log(1 + \exp(\cdot))$, did not capture RGC responses as well as the exponential function (*Pillow et al., 2008*).

## Modeling respones to spatio-temporal stimuli

For spatio-temporal stimuli, the filters for the CBEM and GLM ($\mathbf{k}, \mathbf{k}_e$, and $\mathbf{k}_i$) spanned both space and time. Although the stimulus we used was a $10 \times 10$ grid of pixels, the receptive field (RF) of the neurons did not cover the entire grid. We therefore limited the spatial extent of the linear filters to a $5 \times 5$ grid of pixels, where the center pixel was the strongest point in the GLM stimulus filter.

The filters were represented as a matrix where the columns span the pixel space and the rows span the temporal dimension. The number of parameters was reduced by decomposing the spatio-temporal filters into a low-rank representation (*Pillow et al., 2008*). The filter at pixel $x$ and time $\tau$ became

$$\mathbf{k}(x, \tau) = \sum_{j=1}^{J} \mathbf{k}_{s,j}(x)\mathbf{k}_{t,j}(\tau) \tag{31}$$

where $\mathbf{k}_{s,j}$ was a vector containing the spatial portion of the filter of length 25 (the number of pixels in the RF) and $\mathbf{k}_{t,j}$ represented the temporal portion of the filter. The temporal filters were projected into the same 10-dimensional basis as the temporal filters used to model the full-field stimuli and the spatial filters were represented in the natural pixel basis. For identifiability, we normalized the spatial filters and forced the sign of the center pixel of the spatial filters to be positive. We used rank two filters ($J = 2$) for the CBEM and GLM. Therefore, each filter contained $2 \times 25$ spatial and $2 \times 10$ temporal parameters for a total of 70 parameters. In the GLM, we found no significant improvement using rank three filters. To fit these low-rank filters, we alternated between optimizing over the spatial and temporal components of the filters.

## Evaluating model performance

We evaluated single-trial spike train predictive performance by computing the log-likelihood on a test spike train. We computed the difference between the log (base-2) likelihood under the model and the log-likelihood under a homogeneous rate model ($LL_h$) that captured only the mean spike rate:

$$LL_h = n_{sp} * \log_2(\bar{\lambda}) + (T - n_{sp})\log_2(1 - \bar{\lambda}) \tag{32}$$

$$\bar{\lambda} = \frac{n_{sp}}{T}. \tag{33}$$

where the test stimulus is of length $T$ (in discrete bins) and contains $n_{sp}$ spikes. We then divided by the number of spikes to obtain the predictive performance in units of bits per spike (bits/sp) (*Panzeri et al., 1996*; *Brenner et al., 2000*; *Paninski et al., 2004*)

$$\text{bits per spike} = \frac{LL_{model} - LL_h}{n_{sp}}. \tag{34}$$

We evaluated model predictions of spike rate by simulating 2500 trials from the model for a repeated stimulus. We computed the firing rate, or PSTH, by averaging the number of spikes observed in 1 ms bins and smoothing with a Gaussian filter with a standard deviation of 2 ms. The percent of variance in the PSTH explained by the model is

$$\%\text{variance explained} = 100 \times 1 - \frac{\sum_{t=1}^{T}(PSTH_{data}(t) - PSTH_{model}(t))^2}{\sum_{t=1}^{T}(PSTH_{data}(t) - \overline{PSTH_{data}})^2} \tag{35}$$

where $\overline{PSTH_{data}}$ denotes the average value of the PSTH.

## Acknowledgements

We thank EJ Chichilnisky for generously providing data and valuable discussion. We also thank Il Memming Park and Jacob Yates for helpful comments. This work was supported by the McKnight Foundation (JWP), the Simons Foundation (SCGB AWD1004351, JWP), an NSF CAREER Award IIS-1150186 (JWP), a grant from the NIMH (MH099611, JWP), the Howard Hughes Medical Institute (FR), and a grant from the NIH (EY011850, FR).

## Additional information

### Competing interests

Fred Rieke: Reviewing editor, *eLife*. The other authors declare that no competing interests exist.

### Funding

| Funder | Grant reference number | Author |
| --- | --- | --- |
| McKnight Foundation | | Jonathan W Pillow |
| Simons Foundation | SCGB AWD1004351 | Jonathan W Pillow |
| National Science Foundation | IIS-1150186 | Jonathan W Pillow |
| National Institute of Mental Health | MH099611 | Jonathan W Pillow |
| Howard Hughes Medical Institute | | Fred Rieke |
| National Institutes of Health | EY011850 | Fred Rieke |

The funders had no role in study design, data collection and interpretation, or the decision to submit the work for publication.

### Author contributions

Kenneth W Latimer, Conceptualization, Software, Formal analysis, Validation, Investigation, Visualization, Methodology; Fred Rieke, Conceptualization, Resources, Data curation, Validation, Investigation; Jonathan W Pillow, Conceptualization, Resources, Funding acquisition, Visualization, Methodology, Project administration

### Author ORCIDs

Kenneth W Latimer (iD) https://orcid.org/0000-0002-9981-3903
Fred Rieke (iD) http://orcid.org/0000-0002-1052-2609
Jonathan W Pillow (iD) https://orcid.org/0000-0002-3638-8831

## Ethics

Animal experimentation: Tissue was obtained via the tissue distribution program at the Washington National Primate Research Center. All animal procedures were performed in accordance with IACUC protocols at the University of Washington (IACUC protocol number 4277-01).

## Decision letter and Author response

Decision letter https://doi.org/10.7554/eLife.47012.sa1
Author response https://doi.org/10.7554/eLife.47012.sa2

## Additional files

### Supplementary files

• Transparent reporting form

### Data availability

All modeling tools have been made publicly available at https://github.com/pillowlab/CBEM (copy archived at https://github.com/elifesciences-publications/CBEM). The datasets analyzed in this paper have been previously published as the following: 1. Conductance and cell-attached spike recordings: Philipp Khuc Trong and Fred Rieke (2008). "Origin of correlated activity between parasol retinal ganglion cells." https://doi.org/10.1038/nn.2199. Dataset available via figshare https://figshare.com/articles/ON-Parasol_RGCs_for_the_conductance-based_encoding_model/9636854. 2. Full-field extracellular recordings (including multiple contrasts): VJ Uzzell and EJ Chichilnisky (2004). "Precision of Spike Trains in Primate Retinal Ganglion Cells." https://doi.org/10.1152/jn.01171.2003. Dataset can be accessed through a response to the corresponding author. 3. Spatio-temporal stimuli: Jonathan W Pillow, Jonathon Shlens, Liam Paninski, Alexander Sher, Alan M Litke, EJ Chichilnisky and Eero P Simoncelli (2008). "Spatio-temporal correlations and visual signalling in a complete neuronal population." https://doi.org/10.1038/nature07140. Dataset can be accessed through a response to the corresponding author.

The following previously published dataset was used:

| Author(s) | Year | Dataset title | Dataset URL | Database and Identifier |
|---|---|---|---|---|
| Kenneth W Latimer, Fred Rieke, Jonathan W Pillow | 2019 | ON-Parasol RGCs for the conductance-based encoding model | https://figshare.com/articles/ON-Parasol_RGCs_for_the_conductance-based_encoding_model/9636854 | figshare, 10.6084/m9.figshare.9636854 |

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
