## [Decision Letter]

**Acceptance summary:**

This paper provides a bridge between traditional phenomenological models of neuronal spiking and more biophysically realistic models, by supplying a method to infer excitatory and inhibitory conductances directly from spiking data. This manuscript describes this new model, a conductance-based encoding model (CBEM), and adds a crucial stage in testing this type of biophysical modeling by direct comparison with intracellular data. The CBEM model is validated with spiking and conductance data measured in retinal ganglion cells. Excitatory and inhibitory conductances in both midget and parasol cells from the primate retina can be inferred reliably. The method has potential applications to other cell types beyond the retina and can provide more mechanistic insights into what drives spiking in the brain.

**Decision letter after peer review:**

Thank you for submitting your work entitled "Inferring synaptic inputs from spikes with a conductance-based neural encoding model" for consideration by *eLife*. Your article has been reviewed by three peer reviewers, one of whom is a member of our Board of Reviewing Editors, and the evaluation has been overseen by a Senior Editor. The reviewers have opted to remain anonymous.

All reviewers found this to be an interesting new model, and thought it is a useful tool for the computational neuroscience community to have. The manuscript was well-written and clear, but some major concerns about the broader impacts and applicability of the model were raised, as detailed below in the three individual reviews.

Reviewer #1:

This paper provides a bridge between traditional phenomenological LNP models and more biophysically realistic models, by supplying a method to infer excitatory and inhibitory conductances from spiking data. It is a well-reasoned and well-written account of this new model (CBEM) for computational neuroscience. The CBEM model is validated with a comparison to spiking and conductance data measured in retinal ganglion cells, and compares reasonably well to an LN model fit directly to the conductance data.

1) The biggest concern the current manuscript raises is: what important features of RGC encoding does the current model capture that a GLM cannot? Put another way, the new model explains more of the response variance of RGC cells, but what does that extra variance encode about the stimulus? What does this extra fitting power allow one to explain in terms of RGC computation? Can simulations be added that explore what aspects of stimulus encoding the full CBEM captures better than a GLM?

2) For the checkerboard stimuli (Figure 9), the GLM and CBEM models seem to have fairly comparable fits to the data. If these are the more interesting data to fit, how should one interpret the weaker additional explanatory power of the CBEM?

3) Some more discussion of the results of the model simulations of center-surround stimuli is needed. How do the observed spike rates compare to known RGC responses? The full CBEM model clearly makes different predictions about the spiking response to spatially correlated stimuli (particularly the inhibition of the sustained response). What aspects of RGC computation are captured by the full CBEM model compared to the excitation only CBEM or GLM for these kind of stimuli?

4) Some variations (dynamic gain adaptation, cascade nonlinearities) on LN models for RGC data can explain more of the observed spiking response to natural image sequences, but still fail to predict the majority of response variance. Several groups have shown that deep or recurrent neural nets can be fit to retinal data and explain about 30% more of this variance. How does a CBEM compare? It could potentially have a serious advantage over these NN models, both in the number of parameters and in interpretability. The absence of a fit to more complex stimuli, where LN models are known to fail spectacularly, stands out as a large hole in the current manuscript. Of course the points brought up in comment 1 apply here as well.

Reviewer #2:

This paper introduces a new point-process model applied to retinal ganglion cell data, which is based on model components that infer excitatory and inhibitory conductances to predict spikes. It uses a smorgasbord of data to demonstrate difference successes of this model (better performance over linear models, ability to explain some aspects of contrast adaptation), and has a number of mathematical derivations and some simulations to demonstrate how it works.

Overall, it is a mixture of what appears to be solid methods with some interesting results from combining them. However, I found this paper scattered, flipping between mathematical derivations, modeling results, and findings specific to the retina. This has the effect of not clearly tying down any element convincingly, nor clearly demonstrating novelty – or at least the need for the advances suggested in this paper.

1) If main purpose is to demonstrate a better way to model neurons (or perhaps just retinal ganglion cells), one would want to understand its generality, as well as how it compares to models of similar ilk, particularly phenomenological models of excitation/inhibition such as the Butts, 2011, 2016. There is a lot of overhead in computing the integration-properties of this model, and not clear what is gained over a more phenomenological LNLN cascade model. I see that it is possible there could be a lot to be gained from the CBEM, but the current manuscript does not make this clear.

Furthermore, what is the scope of this model. Do the authors expect this to be a more generally applicable method, or simply a means to model retinal ganglion cell responses? Under what conditions?

2) Relatedly, one wonders how important it is to model the conductances explicitly, versus membrane potential. Comparisons to membrane potential has been explored in several previous modeling approaches requiring only LN modeling (since there will not be two separate terms), including work of Priebe (Mohanty, 2012) in V1 and Demb (Zaghloul, 2005) in retina. More could be said about motivation for wanting to infer conductances past fitting data better. If the motivation is simply to fit data better, see #1 above.

3) The ability to explain adaptation to contrast as total-conductance changes seems interesting, but is only presented to the level of validating the model, and not explored. It is thus difficult to evaluate based on the extensive literature studying this in the retina.

4) Relatedly, the Cui et al., 2016 paper that was cited in this manuscript also posits excitation and inhibition, models recorded synaptic currents and spikes, and offers an explanation for contrast adaptation, but with a different model form. It seems worth more of a discussion here commenting on it. Could their proposed model explain these results? (and/or vice versa). Can one distinguish between their proposed circuit (presynaptic inhibition being dominant) and normal inhibition proposed here? Likewise, although not matching in content as much, Ozuysal and Baccus also model intracellular recordings and contrast adaptation, using yet another model form.

5) The derivations in the subsection “Background: Poisson GLM with spike history”, motivating the modeling, do not make a clear argument that the model goes beyond the phenomenological, given its overly restrictive assumptions (e.g., that E=-I). As soon as the GLM-based biophysical model is derived making explicit assumptions, it seems to say that these assumptions do not hold and defines the CBEM without them. Later data in the paper (and in previous papers) seem to invalidate these assumptions as well. If these assumptions do not hold, what is the purpose of deriving the model in this context?

6) It was very unclear the components of the various models and how they are fit to data, and I could not make sense until poring through the Materials and methods section. This was in part complicated by the use of different models in different figures (fit to different experiments), without a clear overarching structure. It might also be useful to compare and contrast the derived model forms to simpler phenomenological models of excitation/inhibition (see #1 above).

7) The subsection “Capturing spike responses to spatially varying stimuli” based on simulation had very confusing motivation and conclusions, and could be much better fit into the logical structure of the rest of the paper.

Reviewer #3:

The manuscript by Latimer et al., proposes a new model for analyzing single neurons under sensory stimulation. The model framework, here termed conductance-base encoding model (CBEM) is similar in spirit to the widely used generalized linear model (GLM), applying filters to integrate a sensory stimulus and a stochastic process to generate spikes. Also similar to the GLM approach, model parameters can be obtained from experimentally recorded spike trains by a maximum-likelihood approach. A conceptual advance of the CBEM, however, is that the model incorporates separate filters for excitatory and inhibitory inputs and that these inputs are treated as conductance changes of the neuron, providing an additional level of biological realism. The authors validate their approach by analyzing previously published data from parasol retinal ganglion cells recorded in primate retina. They show that the inferred conductances match intracellularly recorded conductances, and they show that their model improves predictions of spike trains to new data as compared to a GLM.

The manuscript presents an interesting and thought-provoking approach. The possibility of inferring features of excitatory and inhibitory input from recorded spike trains may be a great tool for investigating sensory processing. Also, the mathematical connections to the GLM, which are nicely drawn out in the manuscript, provide a good background for understanding how this model framework functions. In the present form, however, one concern is that it remains a bit unclear how directly applicable the model is to other systems or what sort of insights it may provide. I would imagine that a more general discussion of the applicability, limitations, and interpretation of the model, potentially supported by some additional example from data or simulations, would considerably strengthen the manuscript.

Essential revisions:

The successful inference of excitatory and inhibitory inputs is only shown for parasol cells under full-field stimulation. Here, the excitatory and inhibitory filters are nearly inverted with respect to each other, similar to the specific case discussed in the text where the conductance model can be mapped onto a GLM. Therefore, one wonders how the model performs when excitation and inhibition are correlated instead of anti-correlated or when their filters are (nearly) orthogonal to each other. Are correlated components of the inhibition as well recovered as uncorrelated (orthogonal) components, or will there be some bias?

The match of the inferred and measured conductance is impressive and suggests that this should work as a general technique to assess inhibitory input. However, it is not clear to me how the model would treat inhibitory signals that are not direct conductance inputs into the analyzed neuron. In the retina, for example, many circuits are also influenced by presynaptic inhibition, which acts on bipolar cell terminals. Should one expect that such inhibition is captures by the inhibition filter (because it is likely filtered differently than the direct excitatory pathway) or by the excitation filter (because it affects the excitatory conductance)?

Regarding the improvement of response predictions over the GLM, I'm wondering whether the differences in the applied nonlinearities play any role here. While the soft-threshold nonlinearities of the CBEM are backed up nicely by analyses, the exponential nonlinearity of the GLM could create problems in the quantitative predictions, especially for strong activation as is likely the case under full-field stimulation.

To emphasize the significance of the work, maybe the authors could point out more clearly how the method could be used for providing new insight into the investigated neurons. Is there a specific finding regarding the inputs into parasol cells derived from the present analysis that could serve as an example? Or maybe the authors could point out questions where the model framework might provide answers.

[Editors’ note: what now follows is the decision letter after the authors submitted for further consideration.]

Thank you for submitting your article "Inferring synaptic inputs from spikes with a conductance-based neural encoding model" for consideration by *eLife*. Your article has been reviewed by three peer reviewers, one of whom is a member of our Board of Reviewing Editors, and the evaluation has been overseen by Michael Frank as the Senior Editor. The reviewers have opted to remain anonymous.

Essential revisions:

The reviews on this version of the manuscript were mixed, with some definite enthusiasm for the inclusion of the new midget cell data, but some remaining skepticism and questions about the presentation and generality of the current results. Several points need to be fully addressed for this manuscript to be accepted:

1) For the section on interpreting the GLM as a conductance-based model, and the biophysical motivation for the paper:

The model is set up to make some mathematical manipulations in order to remove the voltage dependence of the currents. That wasn't super clear on a first or second pass and should be made more accessible to the reader.

Other points to address for clarity and scholarship:a) There exists a common quasi-biophysical interpretation of the GLM, wherein the output of the linear stage is thought of as an approximation of the intracellular input or voltage. This interpretation requires (1) that E and I inputs are thought of as currents and sum linearly. Near threshold, this approximation might be decent for modeling spiking, as the voltage is ~constant. (2) the integration time of the neuron must be short enough that the response is mostly a function of the inputs, not on its own voltage history. With these constraints presumed satisfied, many previous studies over the years have assumed that the generating function of the GLM has a quasi-biophysical basis. Can this more standard derivation/interpretation be addressed in this manuscript? Does the explicit integration in the "biophysical" GLM detailed in this work (which addresses assumption 2 above) help in fitting neuronal spiking as compared to a standard GLM? No explicit comparisons are made in the manuscript, and should be added.

b) The derivations seem overly long to simply notice that ge(V-Ee) – gi(V-Ei) will have no voltage dependence if ge and gi cancel. This derivation might be difficult to follow for the broad readership of this journal and should be explained more clearly.

c) This should be reorganized so that it doesn't distract from subsequent results. It could be made more clear that the E = -I assumption is made only to connect this to the GLM-like model class, to motivate the model setup, and is relaxed in the subsequent CBEM inference scheme.

2) For the CBEM model setup, justification, and background:a) The GLM derivation is what is used to motivate the CBEM, and excitation and inhibition are rectified (Equation 10 compared with Equation 9). While the motivation for such rectification is that conductances must be non-negative, this seems somewhat misleading. In the retina, rectification of excitation and inhibition (where it exists) often comes from other sources, most notably non-linear synaptic release from bipolar cells (see Schwartz et al., 2012, Turner and Rieke, 2016, and Freeman, 2015), as well as the effect of amacrine cell processing, which often preserve or extend non-linear effects. While conductances cannot be negative (as the authors assert), the rectification in Equation 10 probably has other sources. This issue raises concerns about the motivation for the CBEM and should be addressed and clarified.

b) The model in its current form cannot capture any cell for which either excitation or inhibition is a non-monotonic function of contrast (i.e. any cell with ON-OFF excitation or ON-OFF inhibition). This applies to many other RGC types, and likely most visual neurons downstream in the brain. Please discuss this limitation more fully and argue for the generality of the model. How does this limit the general utility of this approach outside the retina? Can this model be used to infer a larger class of different contrast-response function shapes? That would certainly be impressive, but doesn't appear to be within the reach of the current model.

c) The advances of the CBEM over previous work on models with separate LN components for excitatory and inhibitory inputs needs to be more thoroughly reviewed, placing the CBEM in the context of this work and arguing for its particular advances.

d) Rectification of the inputs (Equation 10) is the basis of the LNLN cascade model used in many papers over the years, starting with the work of Shapley and Victor and Korenberg and Sakai (in 1970s and 80s). It has been explicitly incorporated in likelihood-based models of the retina and LGN in more recent years (Butts et al., 2011) and most recently in (Maheswaranathan et al., 2018). This includes models that explicitly model excitation and inhibition in the retina using nearly equivalent mathematical forms as used in the current manuscript.

The fact that previous models use spike trains to infer excitation and inhibition does seem to detract a bit from the novelty of the CBEM, if the manuscript does not demonstrate why the CBEM's particular form leads to better inference (certainly the validation with intracellular data here is key to drawing these conclusions and is understood as the main innovation in the paper). Answering the following questions would also provide more biological insight into the success of the CBEM: Is the CBEM's ability to match measured excitatory and inhibitory conductances a result of the integration of currents? The rectification of inputs? The difference between conductance and current?

e) What does the restriction that the conductances are non-noisy do to the CBEM in terms of the types of cellular computations it can and cannot reproduce? Are there particular biophysical effects (e.g. stim-dependent spiking noise) that will be missed via this constraint?

3) For a reader interested in applying the method, it will likely be important to get a better feeling for the applicability and interpretability of the data, in particular when no intracellular data are available for comparison.

a) Important questions are, for example, whether the method also works for non-white-noise stimuli and what may be limitations for the applicability of the method, or the interpretation of the obtained filters as corresponding to actual excitatory or inhibitory conductances. If no data are available, simulations and/or thoughtful discussion could help address these concerns.

b) As a specific example, Figure 10 shows that the inhibitory component can capture surround effects, at least for OFF parasol cells. But is it clear that this actually corresponds to inhibition received by the ganglion cell and not a representation of presynaptic inhibition that nonlinearly interacts with excitatory signals? (Presynaptic effects appear to form a major part of the surround in parasol cells, see, e.g., discussion in Turner, Schwartz and Rieke, 2018.)

c) The method currently uses an explicit model and parameters of the output nonlinearity that are obtained from intracellular data. For pure extracellular data, these parameters (or the shape of the nonlinearity) will not be known a priori. How does that affect the model fitting? Can the parameters of the output nonlinearity be included in the fitting procedure?

4) The results concerning contrast adaptation could be shortened or omitted. The CBEM does a bit better than a GLM, but really, both fail at contrast adaptation because one needs to model it explicitly, as has been done in many previous models, including papers cited in this manuscript from the Baccus group.

5) Existing approaches for inferring excitation and inhibition from spike trains are incorrectly labeled in the Introduction as LN modeling, and only briefly mentioned as alternative to the CBEM in the third paragraph of the discussion. This should be corrected, and the CBEM should be presented in this fuller context.

[Editors’ note: further revisions were suggested before acceptance.]

Thank you for resubmitting your work entitled "Inferring synaptic inputs from spikes with a conductance-based neural encoding model" for further consideration at *eLife*. Your revised article has been favorably evaluated by Michael Frank (Senior Editor) and a Reviewing Editor.

The manuscript has been improved but there are some remaining issues that need to be addressed before acceptance, as outlined below:

1) To clarify a bit first: The main model comparison that was sought after by the reviewers had the following aim: Explain how the inference of conductances here (i.e. the biophysical basis of the CBEM) is the key ingredient in successfully predicting the E/I values, as opposed to LNLN models, which have more general nonlinearities (as compared to the GLM's). The reviewers agree with you that the goal here is not to outperform other models per se, in terms of fitting performance or whatnot, but to add interpretable knowledge about the underlying biophysics. The prompt, then, from the reviewers is this: show more clearly and directly how the biophysical assumptions in the CBEM are crucial for getting this E/I estimation right, which allows for the proper interpretation of the model results; show that it's not just the fact that the CBEM (like LNLN models) has a more generalized form of nonlinearity built into it. Essentially, that one needs to model the biophysics in this more correct way to get the right interpretation out.

Here's the concern spelled out more explicitly:

If the CBEM is no better (at predicting the E/I inputs to a cell) than other LNLN models that simply have voltage-like, LN approximations of excitatory and inhibitory inputs, this means that the biophysics presented here is somewhat misleading (at least in the sense that it has to do with considering conductances), which is the current basis of the paper.

Additionally, any more-flexible LNLN models (with many subunits) -- now several in the literature – would actually outperform the CBEM because they can include multiple subunits, and model more general nonlinearities.

It is very possible that the more explicit model of the biophysics in the CBEM would do better and its particular structure is therefore an advance – but the manuscript does not show or address this directly.

There were certainly some things said about this in the response to reviewers, and more of that should be entered into the paper as well.

2) You had a question in the response to reviewers about references for the "quasi-biophysical interpretation of the GLM". The Gerstner references are certainly great here, but please also cite (and if appropriate, discuss) Pillow et al., 2004 and 2005 (where the output of the linear term is explicitly treated as a voltage) and perhaps one of the more recent papers comparing GLM fits to intracellular data.

3) Please add a few more lines to the Discussion section outlining why you expect this model to be of broad utility beyond the retina. Specifically: what sorts of heuristics can a future user of the method employ to decide if the model's assumptions are appropriate for their data? Here, just saying a bit more to justify the breadth of the expected applicability of the model would be sufficient.

---

## [Author Response]

[Editors’ note: the author responses to the first round of peer review follow.]

Reviewer #1:

This paper provides a bridge between traditional phenomenological LNP models and more biophysically realistic models, by supplying a method to infer excitatory and inhibitory conductances from spiking data. It is a well-reasoned and well-written account of this new model (CBEM) for computational neuroscience. The CBEM model is validated with a comparison to spiking and conductance data measured in retinal ganglion cells, and compares reasonably well to an LN model fit directly to the conductance data.1) The biggest concern the current manuscript raises is: what important features of RGC encoding does the current model capture that a GLM cannot? Put another way, the new model explains more of the response variance of RGC cells, but what does that extra variance encode about the stimulus? What does this extra fitting power allow one to explain in terms of RGC computation? Can simulations be added that explore what aspects of stimulus encoding the full CBEM captures better than a GLM?

The CBEM can capture nonlinear response properties that arise from the interaction of excitatory and inhibitory conductance based inputs (which can give rise, for example to gain changes and changes in time constant), whereas the GLM is limited to a single filter, which assumes linear summation of excitatory and inhibitory inputs. In Figure 8 and Figure 10, we presented simulations showing that E‑I dependence leads to different predictions for contrast changes and center surround integration. We will extend these analyses to highlight how the CBEM’s conductances lead to these effects. More broadly, we would like to emphasize that we do not view the paper’s primary contribution as the fact that the CBEM “captures more response variance of RGC cells”. Rather, we think that the fact that it reveals intracellular conductances (verified with intracellular recordings) purely on the basis of spike times is the most important finding, and future application of our model. We will emphasize this primary motivation more clearly in our revision.

2) For the checkerboard stimuli (Figure 9), the GLM and CBEM models seem to have fairly comparable fits to the data. If these are the more interesting data to fit, how should one interpret the weaker additional explanatory power of the CBEM?

For these stimuli, the GLM does a fairly good job of fitting the data, and thus there isn't that much room for improvement. We note that even on natural scene data, the GLM can predict a large amount of the variance (for example, Batty et al., 2017). Despite the similar predictions to white noise, the CBEM predicts significant differences in responses to correlated stimuli.

3) Some more discussion of the results of the model simulations of center-surround stimuli is needed. How do the observed spike rates compare to known RGC responses? The full CBEM model clearly makes different predictions about the spiking response to spatially correlated stimuli (particularly the inhibition of the sustained response). What aspects of RGC computation are captured by the full CBEM model compared to the excitation only CBEM or GLM for these kind of stimuli?

The CBEM captures nonlinear center‑surround modulation that cannot be explained by a single linear filter. Additionally, in our simulations the interactions between inhibitory and excitatory inputs could produce more precise responses than predicted by the GLM or CBEM with only excitation.

4) Some variations (dynamic gain adaptation, cascade nonlinearities) on LN models for RGC data can explain more of the observed spiking response to natural image sequences, but still fail to predict the majority of response variance. Several groups have shown that deep or recurrent neural nets can be fit to retinal data and explain about 30% more of this variance. How does a CBEM compare? It could potentially have a serious advantage over these NN models, both in the number of parameters and in interpretability. The absence of a fit to more complex stimuli, where LN models are known to fail spectacularly, stands out as a large hole in the current manuscript. Of course the points brought up in comment 1 apply here as well.

Again, we apologize that we were not clearer about the primary contributions of our paper. As noted above, our goal is not simply to explain more variance, but to offer biological insights into what drives spiking in retinal ganglion cells. We offered the comparison to LN models and standard GLM as an “added bonus” that the CBEM also can provide a more accurate model of spiking. (Presumably this is because of the higher level of biological realism it achieves). However, the main point of our paper was not to produce a better R^2 value (although a higher R^2 is a consequence of better capturing of the synaptic input to the RGC), but to link purely phenomenological models to biology. We agree of course that applications to natural scenes are an important avenue for future research, and we hope that the CBEM will be provide a tool for studying neural responses obtained from such datasets.

Reviewer #2:

This paper introduces a new point-process model applied to retinal ganglion cell data, which is based on model components that infer excitatory and inhibitory conductances to predict spikes. It uses a smorgasbord of data to demonstrate difference successes of this model (better performance over linear models, ability to explain some aspects of contrast adaptation), and has a number of mathematical derivations and some simulations to demonstrate how it works.Overall, it is a mixture of what appears to be solid methods with some interesting results from combining them. However, I found this paper scattered, flipping between mathematical derivations, modeling results, and findings specific to the retina. This has the effect of not clearly tying down any element convincingly, nor clearly demonstrating novelty – or at least the need for the advances suggested in this paper.1) If main purpose is to demonstrate a better way to model neurons (or perhaps just retinal ganglion cells), one would want to understand its generality, as well as how it compares to models of similar ilk, particularly phenomenological models of excitation/inhibition such as the Butts, 2011, 2016. There is a lot of overhead in computing the integration-properties of this model, and not clear what is gained over a more phenomenological LNLN cascade model. I see that it is possible there could be a lot to be gained from the CBEM, but the current manuscript does not make this clear.Furthermore, what is the scope of this model. Do the authors expect this to be a more generally applicable method, or simply a means to model retinal ganglion cell responses? Under what conditions?

We apologize — the main goal of our model is not a better model of RGCs in terms of spike rate prediction, but to tie phenomenological to biophysical models which include dynamics. The overhead of computing the integration is not actually that big: this is solved with a tri‑diagonal matrix operation which is computed in linear time. What we gain with this is a far more solid connection between our model and biophysical components of the neurons, than is achieved by a phenomenological model. We do however see the merit in including a comparison to the Nonlinear Interaction Model proposed in McFarland, Cui and Butts, (2013).

As with the GLM, we expect that this model may be useful in many sensory regions. The conditions that we expect this model to be useful under conditions that any cascade model might be useful.

2) Relatedly, one wonders how important it is to model the conductances explicitly, versus membrane potential. Comparisons to membrane potential has been explored in several previous modeling approaches requiring only LN modeling (since there will not be two separate terms), including work of Priebe (Mohanty, 2012) in V1 and Demb (Zaghloul, 2005) in retina. More could be said about motivation for wanting to infer conductances past fitting data better. If the motivation is simply to fit data better, see #1 above.

We apologize again for the confusion: we feel the problem of identifying the factors that govern spiking at an intracellular level are worthwhile in their own right (e.g., does an increase in spiking result from a decrease in inhibition or an increase in excitation). So the viewpoint that we take is not that “it is important to model the conductances explicitly in order to obtain a state‑of‑the‑art descriptive model of retinal spike responses”, but rather that identifying the separate excitatory and inhibitory inputs to neurons is an interesting scientific problem in its own right. We did in fact use an LN model, the GLM, as a basis of comparison throughout the paper; this would be practically equivalent to the reviewer’s suggestion of modeling the voltage as LN because our spike rate was a nonlinear function of voltage. There are situations for which the voltage can be modeled as a simple LN model, but such models do not capture cells' responses to all stimuli. We note that of course there are known features of RGC responses that cannot be captured with LN models, as the reviewer has pointed out in comments 1, 3, and 4.

3) The ability to explain adaptation to contrast as total-conductance changes seems interesting, but is only presented to the level of validating the model, and not explored. It is thus difficult to evaluate based on the extensive literature studying this in the retina.

Sorry, we are not sure we understand the reviewer's objection here. Because we are introducing the CBEM and presenting it broadly, we did not aim to explore the entire gamut of contrast adaptation experiments within one subsection of the results. However, we would like to extend the presentation in this section to include information on how the prediction distributions of excitation and inhibition depend on contrast.

4) Relatedly, the Cui et al., 2016 paper that was cited in this manuscript also posits excitation and inhibition, models recorded synaptic currents and spikes, and offers an explanation for contrast adaptation, but with a different model form. It seems worth more of a discussion here commenting on it. Could their proposed model explain these results? (and/or vice versa). Can one distinguish between their proposed circuit (presynaptic inhibition being dominant) and normal inhibition proposed here? Likewise, although not matching in content as much, Ozuysal and Baccus also model intracellular recordings and contrast adaptation, using yet another model form.

Cui et al., model did not define inhibitory and excitatory synaptic conductances in a biophysical model. Comparisons between our two methods is difficult because their work focused on capturing nonlinear effects of presynaptic inhibition in the mouse retina. We feel that exploring these specific nonlinear presynaptic inhibition is beyond the scope of our paper, but it would be interesting to explore in future work. We are familiar with the work of Ozuysal and Baccus, but would like to emphasize that work required intracellular recordings for model fitting — the main achievement of our work is that the model can be used to identify intracellular conductances despite being fit only spike trains; thus it can shed light on underlying biological mechanisms even when only extracellular recordings are available.

5) The derivations in the subsection “Background: Poisson GLM with spike history”, motivating the modeling, do not make a clear argument that the model goes beyond the phenomenological, given its overly restrictive assumptions (e.g., that E=-I). As soon as the GLM-based biophysical model is derived making explicit assumptions, it seems to say that these assumptions do not hold and defines the CBEM without them. Later data in the paper (and in previous papers) seem to invalidate these assumptions as well. If these assumptions do not hold, what is the purpose of deriving the model in this context?

Sorry, but we feel we did not explain this clearly enough. Our point was not to suggest that E=‑I is a good model for real neurons, but to offer a new interpretation for the classic GLM. (This connection to a biophysical model — albeit an unrealistic one — has not to our knowledge been proposed before). The idea of relaxing this overly restrictive constraint is the step that leads to our model, the CBEM.

The main point of our paper is to link phenomenological models to biophysical models. These derivations were only intended show a link between a commonly used statistical model of spiking (GLM) and a biophysical model to demonstrate the assumptions on biophysics placed by the GLM that are not obvious from the purely statistical presentation. We will clarify this motivation in the revision.

6) It was very unclear the components of the various models and how they are fit to data, and I could not make sense until poring through the Materials and methods section. This was in part complicated by the use of different models in different figures (fit to different experiments), without a clear overarching structure. It might also be useful to compare and contrast the derived model forms to simpler phenomenological models of excitation/inhibition (see #1 above).

We apologize for the difficulty in understanding our model. Our motivation for the paper’s organization was to isolate each biophysically interpretable component of the model and relate the component to quantities measures in the real retina. Our Materials and methods section is structured in accordance to each Results section and figure. Secondly, as we mentioned in our response to comment 1, we are willing to compare the CBEM with NIM, a phenomenological model with excitatory and suppressive units.

7) The subsection “Capturing spike responses to spatially varying stimuli” based on simulation had very confusing motivation and conclusions, and could be much better fit into the logical structure of the rest of the paper.

We are sorry this was not clear. The simulation attempts to demonstrate how the models differ in their predictions of center‑surround tuning (with a stimulus that wasn’t used in fitting) despite giving similar fits to noise data. We can provide additional information on the simulation work.

Reviewer #3:[…] The manuscript presents an interesting and thought-provoking approach. The possibility of inferring features of excitatory and inhibitory input from recorded spike trains may be a great tool for investigating sensory processing. Also, the mathematical connections to the GLM, which are nicely drawn out in the manuscript, provide a good background for understanding how this model framework functions. In the present form, however, one concern is that it remains a bit unclear how directly applicable the model is to other systems or what sort of insights it may provide. I would imagine that a more general discussion of the applicability, limitations, and interpretation of the model, potentially supported by some additional example from data or simulations, would considerably strengthen the manuscript.Essential revisions:The successful inference of excitatory and inhibitory inputs is only shown for parasol cells under full-field stimulation. Here, the excitatory and inhibitory filters are nearly inverted with respect to each other, similar to the specific case discussed in the text where the conductance model can be mapped onto a GLM. Therefore, one wonders how the model performs when excitation and inhibition are correlated instead of anti-correlated or when their filters are (nearly) orthogonal to each other. Are correlated components of the inhibition as well recovered as uncorrelated (orthogonal) components, or will there be some bias?

We expect that the specific configuration of excitation and inhibition will matter. However, we have found in simulations that we can recover the conductance filters when inhibition is delayed, but not opposite, to excitation.

The match of the inferred and measured conductance is impressive and suggests that this should work as a general technique to assess inhibitory input. However, it is not clear to me how the model would treat inhibitory signals that are not direct conductance inputs into the analyzed neuron. In the retina, for example, many circuits are also influenced by presynaptic inhibition, which acts on bipolar cell terminals. Should one expect that such inhibition is captures by the inhibition filter (because it is likely filtered differently than the direct excitatory pathway) or by the excitation filter (because it affects the excitatory conductance)?

Presynaptic inhibition is an interesting effect (as noted by reviewer 2 in reference to Cui et al., 2016 which looked at such computations in the mouse retina). However, we believe these effects are beyond the scope of the current paper, but we would like to explore more complicated nonlinear effects in the future.

Regarding the improvement of response predictions over the GLM, I'm wondering whether the differences in the applied nonlinearities play any role here. While the soft-threshold nonlinearities of the CBEM are backed up nicely by analyses, the exponential nonlinearity of the GLM could create problems in the quantitative predictions, especially for strong activation as is likely the case under full-field stimulation.

We found that the GLM performed very poorly with the soft‑rectified nonlinearity for these cells (as was also reported in Pillow et al., 2008). We have made sure to mention this. We also emphasize that the choice of the CBEM’s nonlinearity was by matching voltage to spike rate, and not to maximize the performance of spike rate predictions in response to a stimulus.

To emphasize the significance of the work, maybe the authors could point out more clearly how the method could be used for providing new insight into the investigated neurons. Is there a specific finding regarding the inputs into parasol cells derived from the present analysis that could serve as an example? Or maybe the authors could point out questions where the model framework might provide answers.

Our method provides a quantitative framework to combine different measures of neural activity that cannot be simultaneously measured. In the array recordings we analyzed, we had access to far more data than with intracellular recordings, but we could only observe spikes.

The gain control section of our paper showed that independent excitation and inhibition can capture some of the contrast adaptation observed in RGCs. Because multiple mechanisms contribute to contrast adaptation, this affect could not easily be quantified without a model.

[Editors' note: the author responses to the re-review follow.]

Essential revisions:The reviews on this version of the manuscript were mixed, with some definite enthusiasm for the inclusion of the new midget cell data, but some remaining skepticism and questions about the presentation and generality of the current results. Several points need to be fully addressed for this manuscript to be accepted:1) For the section on interpreting the GLM as a conductance-based model, and the biophysical motivation for the paper:The model is set up to make some mathematical manipulations in order to remove the voltage dependence of the currents. That wasn't super clear on a first or second pass and should be made more accessible to the reader.

We apologize for the confusion. We have revised this section to make clearer that the goal here was to achieve voltage independence. The revised text reads:

“Here we propose a novel biophysically realistic interpretation of the classic Poisson GLM as a dynamical model with conductance-based input. In brief, this involves writing the GLM as a conductance-based model with excitatory and inhibitory conductances governed by affine functions of the stimulus, but constrained so that total conductance is fixed. This removes voltage-dependence of the membrane currents, making the membrane potential itself an affine function of the stimulus.

The remainder of this section lays out the mathematical details of this interpretation explicitly.”

Other points to address for clarity and scholarship:a) There exists a common quasi-biophysical interpretation of the GLM, wherein the output of the linear stage is thought of as an approximation of the intracellular input or voltage. This interpretation requires (1) that E and I inputs are thought of as currents and sum linearly. Near threshold, this approximation might be decent for modeling spiking, as the voltage is ~constant. (2) the integration time of the neuron must be short enough that the response is mostly a function of the inputs, not on its own voltage history. With these constraints presumed satisfied, many previous studies over the years have assumed that the generating function of the GLM has a quasi-biophysical basis. Can this more standard derivation/interpretation be addressed in this manuscript? Does the explicit integration in the "biophysical" GLM detailed in this work (which addresses assumption 2 above) help in fitting neuronal spiking as compared to a standard GLM? No explicit comparisons are made in the manuscript, and should be added.

We thank the reviewers for this comment, and we apologize if we have failed to cite relevant literature about interpretations of the Poisson GLM. We have cited Plesser and Gerstner, 2000 (which first described the “escape rate” approximation to noisy integrate-and-fire as an LN neuron), as well as work from Gerstner on the spike-response model. We are not familiar however with other previous work interpreting the GLM as having a biophysical basis (e.g. in terms of E and I current inputs that sum linearly). If the reviewers wouldn’t mind clarifying which references they are referring to, we would be grateful for the opportunity to add citations and discuss the relationship to previous work more clearly.

We did not fully understand the comment about a lack of explicit comparisons, as we did indeed compare the “conductance based model” (CBEM) to the standard Poisson GLM in Figure7, Figure 8, Figure 10 and Figure 11. Was there another comparison the reviewers would have liked to see? We apologize if we have misunderstood.

We have updated the discussion of these previous interpretations that was included in subsection “Background: Poisson GLM with spike history”. We now state that this interpretation assumes a current-based input rather than conductance based. The biophysical interpretation of the GLM offered in subsection “Interpreting the GLM as a conductance-based model”, unfortunately does not affect the fits data, as it is mathematically equivalent. (We have sought to emphasize that it is a new interpretation, not a new model).

b) The derivations seem overly long to simply notice that ge(V-Ee) – gi(V-Ei) will have no voltage dependence if ge and gi cancel. This derivation might be difficult to follow for the broad readership of this journal and should be explained more clearly.

We thank the reviewers for this comment. We have now added a summary(“In brief,…”, subsection “Interpreting the GLM as a conductance-based model”), which attempts to summarize the idea so that reviewers who are not interested in the detailed derivation can skip the remainder of the section. We have also shortened and simplified the derivation itself, as we agree with the reviewer that the section was overly long; although we still wish to make clear exactly how to relate the GLM filter k_glm defined in the previous section to the linear conductance filter “k” that we introduced in our interpretation. We hope the revised derivation will be simpler and easier to understand.

c) This should be reorganized so that it doesn't distract from subsequent results. It could be made more clear that the E = -I assumption is made only to connect this to the GLM-like model class, to motivate the model setup, and is relaxed in the subsequent CBEM inference scheme.

Thanks for the suggestion. We have reiterated that this assumption is only to construct a GLM at the start of both the subsection “Interpreting the GLM as a conductance-based model” and subsection “The conductance-based encoding model (CBEM)”.

2) For the CBEM model setup, justification, and background:a) The GLM derivation is what is used to motivate the CBEM, and excitation and inhibition are rectified (eq 10 compared with eq 9). While the motivation for such rectification is that conductances must be non-negative, this seems somewhat misleading. In the retina, rectification of excitation and inhibition (where it exists) often comes from other sources, most notably non-linear synaptic release from bipolar cells (see Schwartz et al., 2012, Turner and Rieke, 2016, and Freeman, 2015), as well as the effect of amacrine cell processing, which often preserve or extend non-linear effects. While conductances cannot be negative (as the authors assert), the rectification in Equation 10 probably has other sources. This issue raises concerns about the motivation for the CBEM and should be addressed and clarified.

Thanks for this suggestion. We have clarified that our nonlinearity is primarily aimed at accounting for synaptic thresholding/nonlinearities, instead of just enforcing positive conductances. This is included where we introduce the soft-rectified nonlinearity for RGC data (subsection: Validating the CBEM modeling assumptions with intracellular data”).

b) The model in its current form cannot capture any cell for which either excitation or inhibition is a non-monotonic function of contrast (i.e. any cell with ON-OFF excitation or ON-OFF inhibition). This applies to many other RGC types, and likely most visual neurons downstream in the brain. Please discuss this limitation more fully and argue for the generality of the model. How does this limit the general utility of this approach outside the retina? Can this model be used to infer a larger class of different contrast-response function shapes? That would certainly be impressive, but doesn't appear to be within the reach of the current model.

Our code allows for an arbitrary set of conductances with different nonlinear functions. However, adding more conductances with more nonlinearities makes fitting more challenging. Future work with additional experiments (which are beyond the scope of this current manuscript) would be needed to test how the model can estimate ON-OFF inhibition or excitation. The discussion now reads “For instance, the CBEM's linear-rectified conductance does not capture the non-monotonic portions of the stimulus-conductance nonlinearities observed in the data”

c) The advances of the CBEM over previous work on models with separate LN components for excitatory and inhibitory inputs needs to be more thoroughly reviewed, placing the CBEM in the context of this work and arguing for its particular advances.d) Rectification of the inputs (Equation 10) is the basis of the LNLN cascade model used in many papers over the years, starting with the work of Shapley and Victor and Korenberg and Sakai (in 1970s and 80s). It has been explicitly incorporated in likelihood-based models of the retina and LGN in more recent years (Butts et al., 2011) and most recently in (Maheswaranathan et al., 2018). This includes models that explicitly model excitation and inhibition in the retina using nearly equivalent mathematical forms as used in the current manuscript.The fact that previous models use spike trains to infer excitation and inhibition does seem to detract a bit from the novelty of the CBEM, if the manuscript does not demonstrate why the CBEM's particular form leads to better inference (certainly the validation with intracellular data here is key to drawing these conclusions and is understood as the main innovation in the paper). Answering the following questions would also provide more biological insight into the success of the CBEM: Is the CBEM's ability to match measured excitatory and inhibitory conductances a result of the integration of currents? The rectification of inputs? The difference between conductance and current?

We thank the reviewers for this comment and suggestions which will help make our specific contributions much more clear within the context of existing literature in the revised manuscript. We have provided additional discussion of the relationship between CBEM and previous work with LNLN cascade models, some of which include functionally suppressive units in the Introduction and Discussion section. We have added clarifications to explain that our approach adds on these previous approaches by comparing the model components identified from spike trains to conductances measured in the same cells (Introduction and Discussion section), which to our knowledge has not been performed in previous modeling exercises. Comparing the model’s predictions specifically to excitatory and inhibitory inputs is a primary goal of our work because phenomenological models may combine inputs from amacrine and bipolar cells into common input channels (as described by Maheswaranathan et al., mentioned by the reviewer) and functionally suppressive units may not correspond to the actual inhibition received by the cell. Moreover, our approach takes the additional step of using a biophysical framework, but instead of being defined purely as an LNLN phenomenological model. Thus, our modeling approach may provide a superior tool to combine quantitative biophysical measurements of cells and circuits with measurement of stimulus encoding.

e) What does the restriction that the conductances are non-noisy do to the CBEM in terms of the types of cellular computations it can and cannot reproduce? Are there particular biophysical effects (e.g. stim-dependent spiking noise) that will be missed via this constraint?

This is an excellent point. Without correlations in the two channels, we cannot predict the coding benefits of correlations observed in Cafaro and Rieke, (2010). We have included this in the Discussion section as a possible direction for future research.

3) For a reader interested in applying the method, it will likely be important to get a better feeling for the applicability and interpretability of the data, in particular when no intracellular data are available for comparison.

We thank the reviewers for this suggestion. We have added text discussing the applicability of the CBEM to the Discussion section. Primarily, we believe that the model can be applied when the inputs to a cell are assumed to be approximately linear (or a linearized approximation is assumed through a model; for example, a model V1 as an input to MT).

a) Important questions are, for example, whether the method also works for non-white-noise stimuli and what may be limitations for the applicability of the method, or the interpretation of the obtained filters as corresponding to actual excitatory or inhibitory conductances. If no data are available, simulations and/or thoughtful discussion could help address these concerns.

Thanks for this comment, yes — the method makes no assumptions about Gaussiannity or whiteness of the stimulus. The applicability of our model to non-white noise stimuli is similar to other statistical methods fit by maximum likelihood MAP, because these tools do not rely on STA or STC features. GLMs and neural network models have been successfully fit using naturalistic stimuli.

b) As a specific example, Figure 10 shows that the inhibitory component can capture surround effects, at least for OFF parasol cells. But is it clear that this actually corresponds to inhibition received by the ganglion cell and not a representation of presynaptic inhibition that nonlinearly interacts with excitatory signals? (Presynaptic effects appear to form a major part of the surround in parasol cells, see, e.g., discussion in Turner, Schwartz and Rieke, 2018.)

We appreciate the reviewer’s comment, but we have considered the treatment of pre-synaptic inhibition beyond the scope of this manuscript. However, we are hoping to explore this more thoroughly in future work. Currently, our model only considers independent excitatory and inhibitory signals. We have made sure to better address this point in the Discussion section.

c) The method currently uses an explicit model and parameters of the output nonlinearity that are obtained from intracellular data. For pure extracellular data, these parameters (or the shape of the nonlinearity) will not be known a priori. How does that affect the model fitting? Can the parameters of the output nonlinearity be included in the fitting procedure?

We could fit these parameters in our optimization. We have run some model fits using a wider range of output nonlinearity parameters and found that this did not strongly impact our ability to predict excitatory and inhibitory tuning. Thus, we would caution against making conclusions about the spike nonlinearity that rest on a single setting of those output parameters in the extracellular setting. Instead, we would argue in favor of considering the range output nonlinearity parameters that are consistent with the data (similar to the work of Prinz and Marder), perhaps within a Bayesian framework.

4) The results concerning contrast adaptation could be shortened or omitted. The CBEM does a bit better than a GLM, but really, both fail at contrast adaptation because one needs to model it explicitly, as has been done in many previous models, including papers cited in this manuscript from the Baccus group.

We apologize but we do not fully understand what the reviewer means by the claim that the models fail at contrast adaptation. We appreciate the reviewers’ concerns and we have edited this section to make the limitations of our work more clear. We have better emphasized that we are not attempting to model all aspects of adaptation, but that our test is aimed only to determine to what extent adaptation can be predicted from the fixed LN conductances alone. Additionally, our model does a fair job at predicting gain change across contrast levels. We have added a citation to the recently published modeling work by Ozuysal, Kastner and Baccus, (2018) who concluded that “differences in synaptic threshold in the two pathways” was the primary source of adaptation in their study.

5) Existing approaches for inferring excitation and inhibition from spike trains are incorrectly labeled in the Introduction as LN modeling, and only briefly mentioned as alternative to the CBEM in the third paragraph of the discussion. This should be corrected, and the CBEM should be presented in this fuller context.

We apologize for this oversight, and have rephrased this paragraph to emphasize that we are referring more generally to the cascade model class (to which all these models belong), not just LN. Additionally, we have also elaborated in the final paragraph of the Introduction how the CBEM compares to the cascade models (LNLN models) the reviewer mentions in terms of excitatory and inhibitory conductance estimation.

[Editors' note: further revisions were suggested before acceptance.]

The manuscript has been improved but there are some remaining issues that need to be addressed before acceptance, as outlined below:1) To clarify a bit first: The main model comparison that was sought after by the reviewers had the following aim: Explain how the inference of conductances here (i.e. the biophysical basis of the CBEM) is the key ingredient in successfully predicting the E/I values, as opposed to LNLN models, which have more general nonlinearities (as compared to the GLM's). The reviewers agree with you that the goal here is not to outperform other models per se, in terms of fitting performance or whatnot, but to add interpretable knowledge about the underlying biophysics. The prompt, then, from the reviewers is this: show more clearly and directly how the biophysical assumptions in the CBEM are crucial for getting this E/I estimation right, which allows for the proper interpretation of the model results; show that it's not just the fact that the CBEM (like LNLN models) has a more generalized form of nonlinearity built into it. Essentially, that one needs to model the biophysics in this more correct way to get the right interpretation out.Here's the concern spelled out more explicitly:If the CBEM is no better (at predicting the E/I inputs to a cell) than other LNLN models that simply have voltage-like, LN approximations of excitatory and inhibitory inputs, this means that the biophysics presented here is somewhat misleading (at least in the sense that it has to do with considering conductances), which is the current basis of the paper.Additionally, any more-flexible LNLN models (with many subunits) -- now several in the literature – would actually outperform the CBEM because they can include multiple subunits, and model more general nonlinearities.It is very possible that the more explicit model of the biophysics in the CBEM would do better and its particular structure is therefore an advance – but the manuscript does not show or address this directly.There were certainly some things said about this in the response to reviewers, and more of that should be entered into the paper as well.

We have made clarifications to the Discussion section to clarify that we believe more flexible LNLN models could help in extending the CBEM (see comment 3). The specific aims of the CBEM are primarily that simple biophysical assumptions leads to an accurate prediction of RGC synaptic input tuning with the CBEM (points which have not been directly testing with LNLN models), and we made sure that the paper did not make claims that one must model biophysics.

However, we admit that we are a bit confused by the request to show that the CBEM that our biophysical assumptions are crucial, or that CBEM is better at predicting conductances than other LNLN models. There simply are no models that we are aware of that have sought to predict conductances before!

Note that our paper does *not* represent a classic machine learning approach, where you take a bunch of stimuli and a bunch of voltage data and train a network to approximate the mapping from one to the other. In our paper, we take a stimuli and recorded spike-trains and learn a model to predict the *membrane potential* that underlay those spikes. So to ask, “does it achieve better voltage prediction than an LNLN model that accurately predicts firing rate?” seems like a non non-sequitur to us. It would be like asking how well a network that predicts MNIST digits can predict the labels of ImageNet images. (Such a model can’t, because it doesn’t even have the right output capability to do so!) Similarly, for our model: there are two conductances at any time point (exc and inh) whereas an LNLN model predicting spike rate has only one output.

But please forgive us if we have misunderstood something. Our view is that — while other models might reasonably be adapted to the prediction of conductances from spike train recordings, using either ours or different assumptions about the underlying biology — that is very much hypothetical “future work” that has not yet been carried out, and so is not a benchmark we aim for in this manuscript.

We are grateful for the feedback, nonetheless, as it clarifies some of the confusion the reviewers may have had with our paper. We have tried to revise the Discussion section to clarify that, while more flexible ML approaches may give better spike prediction, our model seeks an explicit link to biological processes which we confirmed using intracellular data (Discussion section). We also suggest some future avenues along which the two approaches (the CBEM and LNLN or deep learning models) might be fruitfully combined to achieve even better models for conductance prediction. (See Discussion section).

2) You had a question in the response to reviewers about references for the "quasi-biophysical interpretation of the GLM". The Gerstner references are certainly great here, but please also cite (and if appropriate, discuss) Pillow et al., 2004 and 2005 (where the output of the linear term is explicitly treated as a voltage) and perhaps one of the more recent papers comparing GLM fits to intracellular data.

Thanks for the suggestion. We have made sure to discuss these references to the stochastic LIF models from Pillow et al., 2004 and 2005 to mention that those references assume the input is a current-based, linear function of the stimulus (subsection “Background: Poisson GLM with spike history”).

3) Please add a few more lines to the Discussion section outlining why you expect this model to be of broad utility beyond the retina. Specifically: what sorts of heuristics can a future user of the method employ to decide if the model's assumptions are appropriate for their data? Here, just saying a bit more to justify the breadth of the expected applicability of the model would be sufficient.

We have included a much more detailed discussion on the utility of the CBEM and provided recommendations for future applications (Discussion section).